# Towards Preventing Global Knowledge Forgetting in Federated Learning with Non-IID Data

**Abhijit Chunduru**[*]                                                            *schunduru@umass.edu*
*University of Massachusetts Amherst*

**Majid Morafah**[*]                                                              *morafahmajid@gmail.com*
*Independent Researcher*

**Mahdi Morafah**                                                        *mmorafah@wharton.upenn.edu*
*The Wharton School of the University of Pennsylvania*

**Vishnu Pandi Chellapandi**                                                          *cvp@purdue.edu*
*Purdue University*

**Ang Li**                                                                        *angliece@umd.edu*
*University of Maryland College Park*

**Reviewed on OpenReview:** *https://openreview.net/forum?id=lhTWPh3Tjm*

## Abstract

Federated learning under client-level data heterogeneity remains challenging despite extensive work on drift correction, regularization, and improved aggregation. In this paper, we argue that an important yet underexplored failure mode is *catastrophic forgetting of the global decision boundary during local training*: as clients optimize their local objectives, they rapidly overfit to client-specific data and erase globally useful multi-class structure, causing server aggregation to average incompatible models rather than accumulate progress. We provide empirical evidence for this phenomenon through a controlled pilot study that directly visualizes decision boundary evolution in federated learning. Our analysis reveals that standard FL methods consistently forget the global decision boundary after local updates, even when clients are initialized from a strong pretrained global model. Motivated by this observation, we propose FEDPROJ, a federated learning framework designed to preserve global functional knowledge throughout local optimization. FEDPROJ maintains a small public-memory buffer and enforces a hard gradient constraint that prevents local updates from increasing a memory-based distillation loss, thereby acting as a safety barrier against global knowledge erosion. At the server, we further employ ensemble distillation on the same public proxy data to consolidate the preserved knowledge into a single global model. We conduct extensive experiments across computer vision and natural language processing benchmarks, covering highly non-IID regimes and domain-shifted settings. The results show that FEDPROJ consistently outperforms state-of-the-art federated learning methods, highlighting the practical importance of explicitly preventing global decision-boundary forgetting.

## 1 Introduction

Federated Learning (FL) McMahan et al. (2017); Kairouz et al. (2021) enables the training of a shared global model across multiple clients without exchanging raw data, making it a widely adopted paradigm for privacy-sensitive and decentralized learning. The canonical FL algorithm, FedAvg McMahan et al. (2017),

---

[*]Equal authorship and contribution.

alternates between local optimization on each client and server-side aggregation of model updates. While FedAvg has demonstrated success in settings with homogeneous data, its performance often degrades substantially in the presence of client-level data heterogeneity—commonly referred to as Non-IID data—leading to slow convergence and poor generalization Glasgow et al. (2022); Woodworth et al. (2020); Li et al. (2019); Chellapandi et al. (2023b;a;c; 2024); Morafah et al. (2024b); Morafah & Morafah (2025); Morafah et al. (2024a).

A large body of prior work has sought to address the challenges posed by Non-IID data through client-drift correction Solans et al. (2024); Li et al. (2020); Karimireddy et al. (2020), improved server-side aggregation and distillation Cheng et al. (2021); Li & Wang (2019), or modified local training strategies Collins et al. (2021). While these approaches improve stability and convergence in many cases, they largely focus on controlling parameter divergence or enhancing aggregation heuristics. In contrast, the *functional consequences* of local training under heterogeneity—namely, what global information is lost during local optimization—remain insufficiently understood.

In this work, we argue that an important and underexplored failure mode in federated learning is *catastrophic forgetting of global knowledge, manifested as the erosion of the global decision boundary during local training.* Under Non-IID data, clients rapidly overfit to their local objectives and overwrite globally useful multi-class structure learned by the aggregated model. As a result, local models become increasingly specialized and mutually incompatible, so that server-side aggregation largely averages divergent solutions rather than compounding progress. Although this phenomenon has been briefly noted in prior work Zhu et al. (2021), a systematic empirical investigation of global knowledge forgetting in the FL literature is lacking.

To make this failure mode explicit, we first conduct a controlled pilot study that directly visualizes decision boundary evolution in federated learning under Non-IID settings. Our findings show that standard FL methods, including FedAvg, consistently forget the global decision boundary after local updates, even when clients are initialized from a strong pretrained global model. This observation reveals that heterogeneity-induced forgetting is not merely an initialization or optimization artifact, but a structural consequence of unconstrained local training.

Motivated by this diagnosis, we propose *FedProj*, a federated learning framework designed to preserve global functional knowledge throughout local optimization. FedProj introduces an explicit constraint on local gradient updates—where forgetting occurs—by leveraging a small public-memory buffer that encodes global knowledge via ensemble predictions. Local updates are projected to prevent increases in a memory-based distillation loss, acting as a hard safety barrier against global knowledge erosion. At the server, we further consolidate the preserved client knowledge through ensemble distillation on the same public proxy data, improving model fusion without relying on stronger assumptions about client similarity.

We evaluate FedProj on a range of computer vision and natural language processing benchmarks, spanning highly Non-IID regimes and domain-shifted settings. Across all experiments, FedProj consistently outperforms state-of-the-art federated learning methods, demonstrating that explicitly preventing global decision-boundary forgetting is both practically important and algorithmically effective.

**Contributions.** Our main contributions are summarized as follows:

- We identify and empirically characterize catastrophic forgetting of global knowledge—specifically, global decision boundary erosion—as a distinct failure mode in federated learning under data heterogeneity.

- We propose *FedProj*, a federated learning method that mitigates global knowledge forgetting by imposing explicit constraints on local gradient updates.

- We provide a convergence analysis showing that FedProj effectively reduces heterogeneity-induced optimization error compared to standard FL methods.

- We conduct extensive experiments on computer vision and natural language processing tasks, demonstrating consistent improvements over state-of-the-art baselines under Non-IID and domain-shifted settings.

**Organization.** The remainder of this paper is organized as follows. Section 2 reviews related work. Section 3 presents the pilot study on global knowledge forgetting. Section 4 introduces the FedProj methodology.

Section 5 details the theoretical results. Section 6 reports experimental results, and Section 7 concludes the paper.

## 2 Related Work

**Continual Learning.** Catastrophic forgetting has been extensively studied in continual learning, where models are trained sequentially on a stream of tasks. A prominent class of approaches mitigates forgetting by constraining parameter updates to preserve previously learned knowledge, often through gradient projection or subspace restriction. Early methods such as Orthogonal Weight Modulation (OWM) and Orthogonal Gradient Descent (OGD) identify directions in parameter space associated with past tasks and restrict new updates to be orthogonal or minimally interfering with them Zeng et al. (2019); Farajtabar et al. (2020). Gradient Projection Memory (GPM) Saha et al. (2021) improves efficiency by maintaining a compact gradient subspace that captures task-relevant directions, while more recent variants relax strict orthogonality to balance stability and plasticity Lin et al. (2022); Saha & Roy (2023).

While these methods provide important inspiration, the continual learning setting differs fundamentally from federated learning. Continual learning methods are designed for sequential task arrival with a single data stream and typically rely on explicit task boundaries or task-specific memory. In contrast, federated learning involves *simultaneous* optimization across multiple clients with heterogeneous data distributions, where forgetting arises not from task transitions but from unconstrained local optimization that overwrites globally useful structure. Moreover, existing gradient-projection methods in continual learning aim to preserve performance on past tasks, whereas our objective is to preserve *global functional knowledge shared across clients* at every communication round. FedProj draws intuition from these works but adapts gradient constraints to the federated setting by defining preservation in terms of ensemble-induced global knowledge rather than task-specific gradients.

**Federated Learning with Non-IID Data.** A large body of work in federated learning addresses data heterogeneity through client drift correction, regularization, and improved aggregation. Methods such as FedProx Li et al. (2020) and Ditto Li et al. (2021b) introduce proximal or personalized objectives to limit deviation from the global model, while others modify aggregation rules to account for heterogeneous local updates (e.g., FedNova Wang et al. (2020)). Another line of work focuses on server-side knowledge distillation and ensemble fusion. For example, FedDF Lin et al. (2020) refines the global model by distilling knowledge from client models, FedET Cho et al. (2022) constructs an ensemble teacher for aggregation, and FedGKT He et al. (2020) transfers knowledge from client models into a larger server model. These approaches improve robustness under heterogeneity, but they operate primarily at the parameter or output-logit level and do not directly intervene in the local optimization process where forgetting occurs.

**Discussion and Positioning.** As a result, existing methods can still allow local updates to drift away from globally useful decision boundaries, since no mechanism explicitly constrains local gradients to preserve global knowledge across classes. Some distillation-based approaches additionally rely on strong assumptions or external resources, such as one-shot distillation with unlabeled public data Gong et al. (2022), without explicitly addressing the dynamics of forgetting during local training. In contrast, our work focuses on a complementary and largely unexplored question: *what global information is lost during local training under data heterogeneity, and how does this loss affect aggregation?* While prior studies primarily analyze convergence behavior or aggregation bias, they do not explicitly characterize the erosion of global decision boundaries induced by local optimization. We address this gap by empirically diagnosing global knowledge forgetting through a controlled pilot study that visualizes decision boundary evolution in federated learning.

Building on this diagnosis, we propose *FedProj*, a federated learning method that explicitly constrains local optimization to preserve global functional knowledge. Rather than modifying the objective with soft regularization terms or relying solely on server-side correction, FedProj imposes a hard gradient-level constraint: local updates are projected to satisfy a global-knowledge half-space constraint induced by a memory-based distillation loss. This directly prevents local gradients from erasing globally shared structure during training. By operating at the level of optimization dynamics rather than parameters or logits alone, FedProj provides a principled mechanism for mitigating catastrophic forgetting in heterogeneous federated learning. Together,

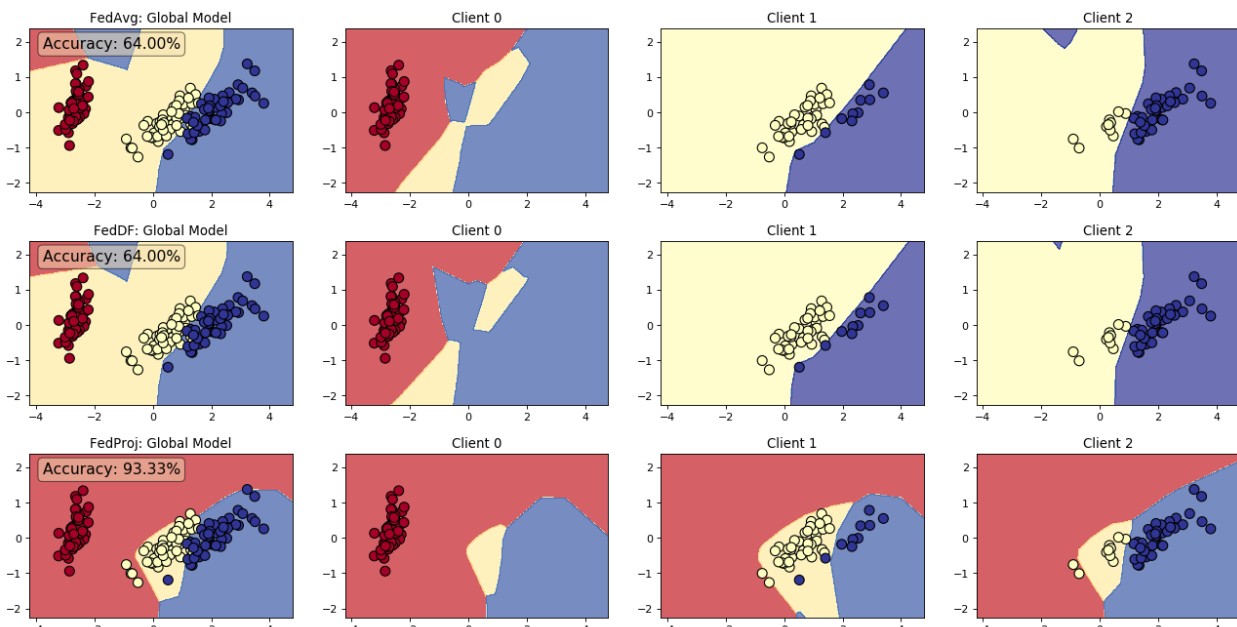

Figure 1: **Visualization of Catastrophic Forgetting of Global Decision Boundaries under Non-IID Federated Learning.** Standard federated learning methods (FedAvg and FedDF) exhibit severe erosion of the global decision boundary after local training on Non-IID client data, resulting in poor global performance (64%). In contrast, *FedProj* preserves the global decision boundary throughout local updates, leading to a substantially more accurate and stable global model (93.33%).

this combination of problem identification, empirical validation, and algorithmic design distinguishes FedProj from existing approaches and reframes catastrophic forgetting as a central challenge in federated learning under data heterogeneity.

## 3 Pilot Study: Diagnosing Global Knowledge Forgetting

This section presents a controlled pilot study designed to *diagnose* a specific failure mode in federated learning under data heterogeneity: catastrophic forgetting of global knowledge during local training. Rather than evaluating overall accuracy alone, our goal is to directly examine how local optimization affects the *global decision boundary* learned by the aggregated model.

**Experimental Setup.** To enable direct visualization of decision boundary dynamics, we use the Iris dataset and apply Principal Component Analysis (PCA) to project the data into two dimensions. The dataset consists of 150 samples from three classes—Setosa, Versicolor, and Virginica—each described by four numerical features. We partition the data across three clients in a Non-IID manner, such that each client predominantly observes samples from a different subset of classes, simulating heterogeneous local objectives. More details on the data partitioning strategy are provided in Appendix C.3.

Each client trains a simple three-layer multi-layer perceptron (MLP). Local optimization is performed using stochastic gradient descent with a learning rate of $1 \times 10^{-3}$ and momentum 0.9. We compare FedAvg, FedDF, and the proposed FedProj method. Training proceeds for 20 communication rounds, with each client performing five local epochs per round. All methods are initialized from the same global model to isolate the effect of local training dynamics.

**Results and Analysis.** Figure 1 visualizes the evolution of class decision boundaries for each method after local training. While the aggregated global model initially exhibits a reasonable multi-class boundary, standard federated learning methods quickly degrade this structure. After local updates, individual client

models diverge sharply from the global decision boundary—most notably for Client 1 and Client 2—indicating that local optimization has overwritten globally shared information in favor of client-specific decision regions.

This divergence highlights a key limitation of existing approaches: although aggregation averages model parameters or predictions, it cannot recover global structure once local training has erased it. As a result, the aggregated model exhibits unstable decision boundaries and substantially reduced accuracy.

In contrast, *FedProj* maintains consistent and coherent decision boundaries across clients throughout local training. By constraining local gradient updates to preserve global functional knowledge, FedProj prevents the erosion of the global decision boundary at its source. Consequently, the aggregated model retains both high accuracy and stable multi-class structure, demonstrating that explicitly preventing global knowledge forgetting during local optimization is critical under data heterogeneity.

## 4 Methodology

We now present *FedProj*, a federated learning algorithm designed to prevent catastrophic forgetting of global knowledge under data heterogeneity. FedProj combines (i) a *client-side gradient projection mechanism* that constrains local optimization dynamics and (ii) a *server-side global knowledge consolidation step* based on ensemble distillation. Together, these components ensure that globally useful functional structure is preserved during local training and coherently fused at the server.

### 4.1 Problem Formulation and Preliminaries

We consider a federated learning system with $N$ clients indexed by $k \in \{1, \ldots, N\}$. Each client $k$ holds a private dataset $\mathcal{D}_k$, and the goal is to learn a shared model parameterized by $\theta \in \mathbb{R}^p$. At communication round $t$, the server broadcasts the current global model $\theta_g^{(t)}$ to a subset of participating clients $\mathcal{S}_t \subseteq \{1, \ldots, N\}$. Each client performs local optimization starting from $\theta_g^{(t)}$, producing updated parameters $\theta_k^{(t+1)}$, which are then aggregated at the server. The standard FedAvg update is given by

$$\theta_g^{(t+1)} \;=\; \sum_{k \in \mathcal{S}_t} \frac{|\mathcal{D}_k|}{\sum_{j \in \mathcal{S}_t} |\mathcal{D}_j|} \, \theta_k^{(t+1)}. \tag{1}$$

While effective under homogeneous data, this procedure degrades under Non-IID client distributions. FedProj addresses this limitation by intervening at two complementary stages:

1. **Client-side gradient projection,** which constrains local updates to preserve global knowledge during optimization;

2. **Server-side global knowledge consolidation,** which distills the preserved client models into a coherent global model.

### 4.2 Local Training with Gradient Projection

**Local Objective.** At round $t$, each client $k \in \mathcal{S}_t$ initializes $\theta_k^{(0)} = \theta_g^{(t)}$ and minimizes its empirical risk

$$\mathcal{L}_{\text{local}}^{(k)}(\theta) \;=\; \frac{1}{|\mathcal{D}_k|} \sum_{(x,y) \in \mathcal{D}_k} \ell(f(x;\theta), y), \tag{2}$$

where $f(x;\theta) \in \mathbb{R}^C$ denotes the model logits and $\ell$ is the cross-entropy loss.

**Global Knowledge Memory.** To prevent local updates from erasing globally shared structure, we introduce a *global knowledge memory* constructed from a small public or proxy dataset $\mathcal{D}_{\text{pub}}$. Let $\mathcal{M} \subset \mathcal{D}_{\text{pub}}$ denote a memory buffer. At round $t$, the server computes ensemble logits

$$Z_{\text{ens}}^{(t)}(x) \;=\; \frac{1}{|\mathcal{S}_t|} \sum_{k \in \mathcal{S}_t} f(x;\theta_k^{(t)}), \qquad x \in \mathcal{M}. \tag{3}$$

These ensemble predictions represent the *global functional knowledge* accumulated across clients. We quantify preservation of this knowledge via the memory loss

$$\mathcal{L}_{\text{mem}}(\theta) \;=\; \frac{1}{|\mathcal{M}|} \sum_{x \in \mathcal{M}} \text{KL}\Big(\sigma\Big(Z_{\text{ens}}^{(t)}(x)\Big) \,\|\, \sigma(f(x;\theta))\Big), \tag{4}$$

where $\sigma(\cdot)$ is the softmax function. Minimizing $\mathcal{L}_{\text{mem}}$ preserves the predictive distribution—and hence the decision boundary—induced by the global ensemble on $\mathcal{M}$.

**Constrained Local Update.** Our objective is to minimize $\mathcal{L}_{\text{local}}^{(k)}$ without increasing $\mathcal{L}_{\text{mem}}$. For a local update $\theta \mapsto \theta'$, we impose the constraint

$$\mathcal{L}_{\text{mem}}(\theta') \;\leq\; \mathcal{L}_{\text{mem}}(\theta). \tag{5}$$

Using a first-order Taylor expansion of $L_{\text{mem}}$ around $\theta$, the constraint is naturally motivated by the condition

$$\langle g_{\text{proj}}, g_{\text{glob}} \rangle \geq 0, \qquad g_{\text{glob}} := \nabla_\theta L_{\text{mem}}(\theta),$$

where $g_{\text{proj}}$ denotes the update direction. Specifically, for a step

$$\theta' = \theta - \eta_{\text{local}} g_{\text{proj}},$$

the descent lemma for an $L_M$-smooth memory loss gives

$$L_{\text{mem}}(\theta') \leq L_{\text{mem}}(\theta) - \eta_{\text{local}} \langle g_{\text{proj}}, g_{\text{glob}} \rangle + \frac{L_M \eta_{\text{local}}^2}{2} \|g_{\text{proj}}\|_2^2.$$

Thus, enforcing $\langle g_{\text{proj}}, g_{\text{glob}} \rangle \geq 0$ provides a *first-order safeguard* against increasing memory loss.

Accordingly, we compute the projected gradient by solving the Euclidean projection problem

$$\min_{g_{\text{proj}}} \frac{1}{2} \|g_{\text{proj}} - g_{\text{new}}\|_2^2 \qquad \text{s.t.} \ \langle g_{\text{proj}}, g_{\text{glob}} \rangle \geq 0, \tag{6}$$

with $g_{\text{new}} = \nabla_\theta L_{\text{local}}^{(k)}(\theta)$. This problem admits the exact closed-form solution

$$g_{\text{proj}} = \begin{cases} g_{\text{new}}, & \langle g_{\text{new}}, g_{\text{glob}} \rangle \geq 0, \\ g_{\text{new}} - \dfrac{\langle g_{\text{new}}, g_{\text{glob}} \rangle}{\|g_{\text{glob}}\|_2^2} g_{\text{glob}}, & \langle g_{\text{new}}, g_{\text{glob}} \rangle < 0 \text{ and } \|g_{\text{glob}}\|_2 > 0. \end{cases} \tag{7}$$

If $\|g_{\text{glob}}\|_2^2$ is too small numerically, we skip projection in implementation, since the memory signal is then too weak to define a stable correction. The client update is

$$\theta \leftarrow \theta - \eta_{\text{local}} g_{\text{proj}}. \tag{8}$$

This projection enforces a global-knowledge half-space constraint at the level of first-order local optimization: local training proceeds along the closest feasible direction relative to the unconstrained local gradient.

### 4.3 Server-Side Global Knowledge Consolidation

After local updates, the server aggregates the client models. Rather than relying solely on parameter averaging, FedProj performs *global knowledge consolidation* via ensemble distillation.

The server initializes a student model $\theta_g$ using the FedAvg aggregate and refines it by minimizing

$$\mathcal{L}_{\text{KD}}(\theta_g) \;=\; \mathbb{E}_{x \sim \mathcal{D}_{\text{pub}}} \text{KL}\Big(\sigma\Big(Z_{\text{ens}}^{(t+1)}(x)\Big) \,\|\, \sigma(f(x;\theta_g))\Big), \tag{9}$$

where $Z_{\text{ens}}^{(t+1)}$ is the ensemble of post-update client models. This step consolidates the preserved global knowledge into a single coherent global model.

**Discussion.** Client-side projection prevents catastrophic forgetting at its source—local optimization—while server-side distillation consolidates complementary knowledge across heterogeneous clients. Without projection, distillation must reconcile inconsistent models; without consolidation, projection alone cannot fully exploit client diversity. Their combination yields a stable and effective federated learning procedure under Non-IID data.

## 5 Theory

We provide two theoretical results. First, we formalize how preserving global functional knowledge on the public memory buffer implies preservation of the global decision boundary on that buffer. Second, we give a convergence guarantee showing that FedProj reduces the *effective heterogeneity* of local updates under a mild alignment condition between the public-memory gradient and the true global gradient.

### 5.1 Global Knowledge Preservation Implies Label Stability on the Buffer

We model the decision boundary as a property of the model's *function* rather than its parameters.

**Definition 5.1** (Global functional knowledge on a buffer). Let $f(\cdot; \theta) \in \mathbb{R}^C$ be logits and let $\mathcal{M} \subset \mathcal{D}_{\text{pub}}$ be the memory buffer. The global functional knowledge induced by $\theta$ on $\mathcal{M}$ is the collection $\{f(x; \theta)\}_{x \in \mathcal{M}}$ (equivalently $\{\sigma(f(x; \theta))\}_{x \in \mathcal{M}}$).

**Definition 5.2** (Decision boundary restricted to $\mathcal{M}$). For $x \in \mathcal{M}$, define the predicted label $y_\theta(x) = \arg\max_j f_j(x; \theta)$ and the (multi-class) margin

$$m_\theta(x) = f_{y_\theta(x)}(x; \theta) - \max_{j \neq y_\theta(x)} f_j(x; \theta).$$

The decision boundary restricted to $\mathcal{M}$ is the set of low-margin points $\mathcal{B}_\theta(\mathcal{M}) = \{x \in \mathcal{M} : m_\theta(x) \approx 0\}$.

**Proposition 5.3** (Label stability under memory-loss control on the buffer). *Let $\theta'$ be a local update from $\theta$, and define the per-sample KL on the memory buffer $M$ by*

$$\mathrm{KL}_x(\theta, \theta') := \mathrm{KL}(\sigma(f(x; \theta)) \,\|\, \sigma(f(x; \theta'))).$$

*For $x \in M$, let*

$$p(x) := \sigma(f(x; \theta)), \qquad y_\theta(x) := \arg\max_j p_j(x),$$

*and define the predictive probability gap*

$$\Delta_\theta(x) := p_{y_\theta(x)}(x) - \max_{j \neq y_\theta(x)} p_j(x).$$

*Then, for any $x \in M$,*

$$\|\sigma(f(x; \theta')) - \sigma(f(x; \theta))\|_1 \leq \sqrt{2\,\mathrm{KL}_x(\theta, \theta')} \qquad \textit{(Pinsker).}$$

*In particular, if*

$$\max_{x \in M} \mathrm{KL}_x(\theta, \theta') \leq \delta \qquad \textit{and} \qquad \Delta_\theta(x) \geq \gamma_p > 0 \;\; \forall x \in M,$$

*then whenever*

$$\delta < \frac{\gamma_p^2}{8},$$

*we have*

$$y_{\theta'}(x) = y_\theta(x) \qquad \forall x \in M.$$

*That is, sufficiently small KL shift preserves the induced classification on the memory buffer.*

Proposition 5.3 clarifies the relationship between our empirical motivation (global decision-boundary forgetting) and the memory-based objective used by FedProj. Controlling the KL-based memory loss controls the predictive distribution on the proxy buffer $M$, and sufficiently small predictive drift preserves the induced classification on that buffer. Importantly, this statement is *local to the support covered by $M$*: by itself it does not imply preservation on the full client or test distribution. Extending the guarantee beyond $M$ requires an additional coverage / representativeness assumption relating the proxy buffer to the target distribution. In practice, we therefore interpret $L_{\mathrm{mem}}$ as a surrogate functional objective whose usefulness depends on the quality and diversity of the proxy buffer.

## 5.2 Convergence and Reduced Effective Heterogeneity

We analyze FedProj under standard smooth nonconvex assumptions used in federated optimization.

Let the global objective be $F(\theta) = \sum_{k=1}^{N} p_k F_k(\theta)$ where $F_k(\theta) = \mathbb{E}_{(x,y)\sim\mathcal{D}_k}\ell(f(x;\theta),y)$ and $\sum_k p_k = 1$. Define the heterogeneity at $\theta$ by

$$\zeta^2(\theta) := \sum_{k=1}^{N} p_k \big\|\nabla F_k(\theta) - \nabla F(\theta)\big\|_2^2, \qquad \zeta^2 := \sup_\theta \zeta^2(\theta).$$

FedProj uses the projected local direction $g_{\mathrm{proj}}^{(k)}$ obtained from equation 6.

**Assumption 5.4** (Smoothness and bounded stochastic noise)**.** *Each $F_k$ is $L$-smooth. Client stochastic gradients are unbiased and have bounded variance: $\mathbb{E}[g^{(k)}(\theta)] = \nabla F_k(\theta)$ and $\mathbb{E}\|g^{(k)}(\theta) - \nabla F_k(\theta)\|_2^2 \leq \sigma^2$.*

**Assumption 5.5** (Public-memory alignment)**.** *Let $g_{\mathrm{glob}}(\theta) = \nabla_\theta \mathcal{L}_{\mathrm{mem}}(\theta)$ be the memory-loss gradient computed on $\mathcal{M}$ using the current server ensemble. Assume that, for all $\theta$ encountered by the algorithm, $g_{\mathrm{glob}}(\theta)$ is a descent direction for the global objective in the sense that*

$$\langle \nabla F(\theta), g_{\mathrm{glob}}(\theta)\rangle \geq \kappa \|\nabla F(\theta)\|_2 \|g_{\mathrm{glob}}(\theta)\|_2 \quad \textit{for some } \kappa \in (0,1].$$

**Theorem 5.6** (FedProj provides a non-expansive gradient guarantee and converges to a stationary point)**.** *Under Assumptions 5.4–5.5, consider $T$ communication rounds with $E$ local steps per participating client and server aggregation by weighted averaging. For a sufficiently small stepsize $\eta_{\mathrm{local}} \leq \frac{1}{cLE}$ (for an absolute constant $c$), the FedProj iterates satisfy*

$$\frac{1}{T}\sum_{t=0}^{T-1}\mathbb{E}\big[\|\nabla F(\theta_g^{(t)})\|_2^2\big] \leq \mathcal{O}\left(\frac{F(\theta_g^{(0)}) - F^\star}{\eta_{\mathrm{local}}TE}\right) + \mathcal{O}\left(\eta_{\mathrm{local}}L\left(\sigma^2 + \frac{\zeta^2}{\kappa}\right)\right),$$

*where $F^\star = \inf_\theta F(\theta)$.*

Theorem 5.6 provides a nonconvex convergence guarantee for FedProj under the stated assumptions.

# 6 Experiments

## 6.1 Main Experimental Setup

**Datasets and Architectures.** We evaluate FEDPROJ on both computer vision (CV) and natural language processing (NLP) tasks. For CV, we consider image classification on CIFAR-10/100 Krizhevsky et al. (2009) and CINIC-10 Darlow et al. (2018). For NLP, we fine-tune pretrained language models on MNLI Williams et al. (2017), SST-2 Socher et al. (2013), and MARC Keung et al. (2020). We use ResNet-8 for CIFAR-10, ResNet-18 for CIFAR-100 and CINIC-10, and TinyBERT Jiao et al. (2019) for all NLP tasks. To simulate statistical heterogeneity, we partition data across clients using a Dirichlet prior Hsu et al. (2019) with concentration $\beta \in \{0.3, 0.5\}$, where smaller $\beta$ induces more severe label skew.

**Federated Learning Protocol.** For CV tasks, we use 100 clients for CIFAR-10 and CINIC-10, and 50 clients for CIFAR-100, with a fixed client sampling rate of 10% per communication round. We train for 100

rounds on CIFAR-10/100 and 60 rounds on CINIC-10, using 20 local epochs per round for all CV experiments. For NLP tasks, we use 15 clients with a 30% sampling rate, 1 local epoch, and 15 communication rounds.

To implement the public-memory and distillation components, we use auxiliary public datasets that are disjoint from the private client data: CIFAR-100 for CIFAR-10/CINIC-10, ImageNet-100 Deng et al. (2009) for CIFAR-100, SNLI Bowman et al. (2015) for MNLI, Sentiment140 Go et al. (2009) for SST-2, and Yelp Zhang et al. (2015) for MARC. This choice reflects the common FL assumption that a small proxy dataset is available for calibration or distillation, without requiring access to private data.

**Implementation Details.** We implement all methods in PyTorch 2.4.1 and run experiments on NVIDIA RTX 3090 GPUs using the FedZoo benchmark Morafah et al. (2023). Our anonymous implementation is available at `https://github.com/Abhijit4-debug/FedProj_TMLR`. We use Adam with learning rate $10^{-3}$ for CV and $3 \times 10^{-5}$ for NLP tasks. Server-side distillation uses KL divergence with temperature $T = 3$, performed for 1 epoch on CIFAR-10, CINIC-10, and all NLP tasks, and for 3 epochs on CIFAR-100. We use batch sizes 256 (CV) and 128 (NLP).

**Baselines and Evaluation.** We compare FEDPROJ against representative baselines spanning optimization, regularization, and distillation families: FEDAVG McMahan et al. (2017), FEDPROX Li et al. (2020), FEDNOVA Wang et al. (2020), FEDDF Lin et al. (2020), FEDET Cho et al. (2022), MOON Li et al. (2021a), FEDDYN Acar et al. (2021), and FEDRCL Seo et al. (2024). We report mean and standard deviation over three independent runs with different random seeds, evaluated by global test accuracy of the final global model.

## 6.2 Main Experimental Results

**Performance on CV Tasks.** Table 1 reports results on CIFAR-10/100 and CINIC-10 under two heterogeneity levels. Across all datasets and both Dirichlet settings, FEDPROJ achieves the best performance, reaching 65.52% and 69.88% on CIFAR-10, 35.27% and 38.06% on CIFAR-100, and 41.46% and 41.63% on CINIC-10 (for $\beta = 0.3$ and $\beta = 0.5$, respectively). Notably, the gains are more pronounced under stronger heterogeneity ($\beta = 0.3$), consistent with our diagnosis that local training increasingly deviates from globally useful decision boundaries as label skew intensifies.

Among baselines, MOON performs competitively under moderate heterogeneity ($\beta = 0.5$) but degrades substantially under stronger skew, highlighting the brittleness of representation-regularization when client objectives become highly misaligned. On CIFAR-100, which is both higher-dimensional and fine-grained, optimization-centered methods (FedNova, FedDyn) exhibit lower accuracy and higher variance, suggesting that correcting for client drift alone does not fully address the loss of global functional structure. On CINIC-10, which introduces additional domain shift, FEDPROJ improves over distillation-based methods (FedDF, FedET), indicating that server-side distillation is most effective when the client models remain functionally aligned rather than having diverged during local optimization. Overall, these results support the central mechanism of FEDPROJ: constraining local updates to preserve global knowledge improves both robustness to heterogeneity and the quality of subsequent server-side consolidation.

Table 1: **Performance on CIFAR-10, CIFAR-100, and CINIC-10** under Dirichlet($\beta = 0.3, 0.5$).

| Baseline | CIFAR-10 | | CIFAR-100 | | CINIC-10 | |
|---|---|---|---|---|---|---|
| | **Dir($\beta$=0.3)** | **Dir($\beta$=0.5)** | **Dir($\beta$=0.3)** | **Dir($\beta$=0.5)** | **Dir($\beta$=0.3)** | **Dir($\beta$=0.5)** |
| FedAvg McMahan et al. (2017) | $63.19 \pm 1.48$ | $66.41 \pm 0.56$ | $33.72 \pm 0.17$ | $37.18 \pm 0.09$ | $40.59 \pm 0.12$ | $40.70 \pm 0.18$ |
| FedProx Li et al. (2020) | $61.40 \pm 0.92$ | $67.34 \pm 0.36$ | $33.96 \pm 0.88$ | $36.66 \pm 0.49$ | $40.69 \pm 0.07$ | $40.80 \pm 0.17$ |
| FedNova Wang et al. (2020) | $63.43 \pm 0.99$ | $67.93 \pm 0.49$ | $33.40 \pm 0.55$ | $36.40 \pm 0.48$ | $39.81 \pm 0.25$ | $40.01 \pm 0.18$ |
| FedDyn Acar et al. (2021) | $63.35 \pm 1.03$ | $67.53 \pm 0.71$ | $33.61 \pm 0.36$ | $36.52 \pm 0.39$ | $40.43 \pm 0.11$ | $40.59 \pm 0.10$ |
| MOON Li et al. (2021a) | $61.09 \pm 1.36$ | $68.83 \pm 0.78$ | $30.29 \pm 0.71$ | $34.49 \pm 0.09$ | $40.64 \pm 0.10$ | $40.79 \pm 0.14$ |
| FedRCL Seo et al. (2024) | $62.14 \pm 0.51$ | $67.26 \pm 0.87$ | $33.91 \pm 0.20$ | $36.77 \pm 0.11$ | $40.72 \pm 0.08$ | $40.81 \pm 0.16$ |
| FedDF Lin et al. (2020) | $63.92 \pm 2.02$ | $68.04 \pm 0.83$ | $33.65 \pm 0.65$ | $36.60 \pm 0.30$ | $40.64 \pm 0.37$ | $40.77 \pm 0.11$ |
| FedET Cho et al. (2022) | $62.79 \pm 1.20$ | $66.46 \pm 0.33$ | $32.85 \pm 0.31$ | $36.21 \pm 0.14$ | $39.11 \pm 0.21$ | $39.20 \pm 0.12$ |
| **FedProj** | $\mathbf{65.52} \pm 0.86$ | $\mathbf{69.88} \pm 0.03$ | $\mathbf{35.27} \pm 0.11$ | $\mathbf{38.06} \pm 0.21$ | $\mathbf{41.46} \pm 0.55$ | $\mathbf{41.63} \pm 0.21$ |

Table 2: **Performance Results for NLP Task on MNLI, SST-2 and MARC.**

| Private | Public | Baseline | Dir($\beta$=0.3) | Dir($\beta$=0.5) |
|---|---|---|---|---|
| MNLI Williams et al. (2017) | SNLI Bowman et al. (2015) | FedAvg | 35.67$\pm$1.21 | 41.91$\pm$3.98 |
| | | FedDF | 36.65$\pm$1.32 | 41.07$\pm$5.88 |
| | | FedET | 36.10$\pm$3.34 | 36.50$\pm$3.39 |
| | | FedProj | **44.38**$\pm$3.91 | **45.13**$\pm$3.10 |
| SST2 Socher et al. (2013) | Sent140 Go et al. (2009) | FedAvg | 56.96$\pm$1.36 | 55.08$\pm$6.46 |
| | | FedDF | 51.43$\pm$2.19 | 54.45$\pm$3.86 |
| | | FedET | 54.96$\pm$8.31 | 56.36$\pm$9.42 |
| | | FedProj | **64.80**$\pm$5.1 | **65.98**$\pm$2.87 |
| MARC Keung et al. (2020) | Yelp Zhang et al. (2015) | FedAvg | 37.21$\pm$2.85 | 40.86$\pm$2.89 |
| | | FedDF | 40.74$\pm$2.91 | 38.40$\pm$6.05 |
| | | FedET | 37.02$\pm$3.39 | 40.05$\pm$2.94 |
| | | FedProj | **45.15**$\pm$1.59 | **46.52**$\pm$4.42 |

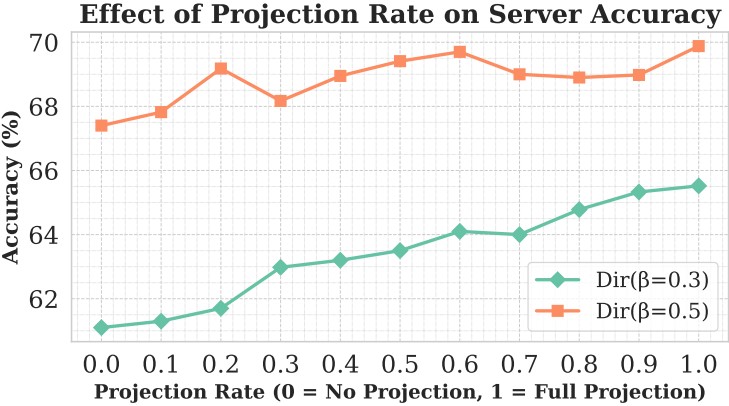

Figure 2: Impact of Projection Dropout.

**Performance on NLP Task.** Table 2 presents results on three NLP private-public dataset pairs: MNLI–SNLI, SST-2–Sentiment140, and MARC–Yelp, under two Dirichlet non-IID settings (Dir($\beta$=0.3) and Dir($\beta$=0.5)). Similar to CV results, on NLP experiments FedProj consistently outperforms existing baselines, demonstrating superior robustness across heterogeneous client scenarios. On MNLI( Williams et al. (2017)), paired with SNLI( Bowman et al. (2015)), FedProj delivers substantial gains. Under Dir($\beta$=0.3), it achieves 44.38 accuracy, surpassing FedDF (36.65) and FedET (36.10), with an improvement exceeding 7% compared to FedAvg. These results expose the limitations of standard aggregation approaches under skewed data distributions. FedET underperforms FedAvg, revealing the downside of relying on uncertainty estimates from pretrained language models, which produce overconfident and poorly calibrated predictions( Wang et al. (2022); Guo et al. (2017); Xiao et al. (2022)). For SST-2( Socher et al. (2013))–Sentiment140( Go et al. (2009)), FedProj delivers state-of-the-art accuracy across both settings, improving by nearly 9% over the strongest baseline. This demonstrates FedProj's effectiveness in bridging domain gaps between curated sentiment labels and noisy social media text. On MARC( Keung et al. (2020))–Yelp( Zhang et al. (2015)), involving multilingual and domain-diverse reviews, FedProj maintains its dominance with 4–6% gains across both non-IID scenarios. These results establish FedProj's effectiveness against both linguistic and domain shifts in realistic federated NLP applications.

### 6.3 Ablation Studies

**Impact of Gradient Projection.** To isolate the effect of the client-side constraint, we ablate projection on CIFAR-10 by randomly omitting the projection step during local updates (Fig. 2). Under strong heterogeneity (Dir($\beta = 0.3$)), accuracy increases monotonically with the projection rate, with full projection achieving

the best performance; even moderate removal sharply degrades accuracy, often reverting toward FEDAVG-level performance. Under milder heterogeneity ($\text{Dir}(\beta = 0.5)$), performance remains stable for moderate projection rates but drops once projection is mostly removed, suggesting that the constraint becomes critical precisely when client objectives are strongly misaligned.

In addition, removing the weight divergence term (Table 8) has only a marginal effect, whereas removing projection substantially degrades performance. Moreover, the consistent gains of FEDPROJ over FEDDF and FEDET—which also employ server-side distillation—further indicate that projection, rather than distillation alone, is the primary driver of improvement. Together, these ablations support the core claim of the paper: explicitly constraining local optimization to preserve global knowledge is essential for preventing catastrophic boundary erosion under heterogeneous federated learning.

## 7 Conclusion and Future Work

In this work, we identify *catastrophic forgetting of global decision boundaries* as a distinct and practically important failure mode in federated learning under data heterogeneity. Through a controlled pilot study, we show that standard local training can rapidly overwrite globally useful functional structure, causing client models to drift toward locally optimal boundaries and undermining subsequent aggregation.

Motivated by this diagnosis, we propose FEDPROJ, a federated learning framework that explicitly preserves global functional knowledge during local optimization and then consolidates it at the server. FEDPROJ combines: (i) a client-side gradient projection mechanism induced by a memory-based distillation loss, which acts as a *first-order safeguard* against local updates that conflict with the proxy global memory objective, and (ii) a server-side ensemble distillation step that consolidates client knowledge into a single global model using a small public proxy dataset. Across a range of CV and NLP benchmarks under realistic Non-IID and domain-shift settings, FEDPROJ consistently improves accuracy and robustness over strong baselines, highlighting the benefits of intervening directly on local optimization dynamics rather than relying solely on soft regularization or aggregation heuristics. These findings are further supported by extensive hyperparameter sensitivity analysis and ablation studies (Appendix C.5), confirming that the reported gains are robust across a wide range of local epochs, buffer sizes, and regularization coefficients, and are not artifacts of any particular configuration.

More broadly, our results suggest that preserving global functional structure during local training is a useful design principle for federated learning under heterogeneity. At the same time, several important directions remain open. First, like other public-data/distillation-based FL methods, FEDPROJ relies on a proxy memory buffer at the server; while our ablations and cross-domain experiments suggest robustness to moderate buffer-size variation and moderate public/private mismatch, understanding the limits of this robustness under severe coverage gaps or extreme domain shift remains an important direction for future work. Second, our current theory establishes buffer-level functional preservation and non-expansive geometric properties under explicit assumptions; tightening the bridge between proxy-buffer preservation and full target-distribution generalization is a natural next step. Finally, an especially promising extension is to study how global-knowledge-preserving gradient constraints interact with minority-client and rare-class representation in federated learning, since preventing majority-driven overwriting of shared structure may also help improve robustness and fairness for underrepresented groups.

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

# A   Additional Theory and Proofs

## A.1   Proof of Boundary Stability Result

We first formalize the relationship between KL control and stability of the predicted label on the buffer.

**Lemma A.1** (Pinsker bound for per-sample predictive shift). *For any $x$ and any $\theta, \theta'$, let $p = \sigma(f(x;\theta))$ and $q = \sigma(f(x;\theta'))$. Then*

$$\|p - q\|_1 \leq \sqrt{2\,\mathrm{KL}(p\|q)}.$$

*Proof.* This is Pinsker's inequality. ☐

**Lemma A.2** (Top-1 label stability under small predictive shift). *Fix $x$ and let $p, q \in \Delta^{C-1}$ be two predictive distributions, with*

$$y = \arg\max_j p_j, \qquad \Delta_p := p_y - \max_{j \neq y} p_j.$$

*If*

$$\|p - q\|_1 < \frac{\Delta_p}{2},$$

*then*

$$\arg\max_j q_j = y.$$

*Proof.* For the winning class $y$,

$$q_y \geq p_y - \|p - q\|_\infty \geq p_y - \|p - q\|_1.$$

For any $j \neq y$,

$$q_j \leq p_j + \|p - q\|_\infty \leq p_j + \|p - q\|_1.$$

Therefore,

$$q_y - q_j \geq (p_y - p_j) - 2\|p - q\|_1 \geq \Delta_p - 2\|p - q\|_1.$$

If $\|p - q\|_1 < \Delta_p/2$, then $q_y > q_j$ for all $j \neq y$, which proves the claim. ☐

*Proof of Proposition 5.3.* Fix $x \in M$ and write

$$p(x) = \sigma(f(x;\theta)), \qquad q(x) = \sigma(f(x;\theta')).$$

By Lemma A.1,

$$\|p(x) - q(x)\|_1 \leq \sqrt{2\,\mathrm{KL}_x(\theta, \theta')}.$$

If $\mathrm{KL}_x(\theta, \theta') \leq \delta$ for all $x \in M$, then

$$\|p(x) - q(x)\|_1 \leq \sqrt{2\delta} \qquad \forall x \in M.$$

Now assume that $\Delta_\theta(x) \geq \gamma_p > 0$ for all $x \in M$. If

$$\delta < \frac{\gamma_p^2}{8},$$

then

$$\sqrt{2\delta} < \frac{\gamma_p}{2} \leq \frac{\Delta_\theta(x)}{2} \qquad \forall x \in M.$$

Applying Lemma A.2 pointwise shows that

$$y_{\theta'}(x) = y_\theta(x) \qquad \forall x \in M.$$

Hence the induced classification on the memory buffer is preserved. ☐

## A.2 Projection Properties: Feasibility and Minimal Intervention

**Proposition A.3** (Feasibility and exact closed-form projection). *Assume $\|g_{\text{glob}}\|_2 > 0$. The solution to*

$$\min_{g_{\text{proj}}} \frac{1}{2}\|g_{\text{proj}} - g_{\text{new}}\|_2^2 \qquad s.t. \ \langle g_{\text{proj}}, g_{\text{glob}} \rangle \geq 0$$

*is*

$$g_{\text{proj}} = \begin{cases} g_{\text{new}}, & \langle g_{\text{new}}, g_{\text{glob}} \rangle \geq 0, \\ g_{\text{new}} - \dfrac{\langle g_{\text{new}}, g_{\text{glob}} \rangle}{\|g_{\text{glob}}\|_2^2} \, g_{\text{glob}}, & \langle g_{\text{new}}, g_{\text{glob}} \rangle < 0, \end{cases}$$

*and satisfies*

$$\langle g_{\text{proj}}, g_{\text{glob}} \rangle \geq 0,$$

*with equality in the active case.*

*Proof.* This is a one-constraint convex quadratic program. Let

$$\mathcal{L}(g_{\text{proj}}, \lambda) = \frac{1}{2}\|g_{\text{proj}} - g_{\text{new}}\|_2^2 - \lambda \langle g_{\text{proj}}, g_{\text{glob}} \rangle, \qquad \lambda \geq 0.$$

The KKT conditions are

$$g_{\text{proj}} - g_{\text{new}} - \lambda g_{\text{glob}} = 0, \qquad \lambda \geq 0, \qquad \lambda \langle g_{\text{proj}}, g_{\text{glob}} \rangle = 0, \qquad \langle g_{\text{proj}}, g_{\text{glob}} \rangle \geq 0.$$

Thus

$$g_{\text{proj}} = g_{\text{new}} + \lambda g_{\text{glob}}.$$

If $\langle g_{\text{new}}, g_{\text{glob}} \rangle \geq 0$, then $\lambda = 0$ and $g_{\text{proj}} = g_{\text{new}}$ is feasible and optimal. Otherwise, the constraint is active, so

$$\langle g_{\text{proj}}, g_{\text{glob}} \rangle = 0.$$

Substituting $g_{\text{proj}} = g_{\text{new}} + \lambda g_{\text{glob}}$ gives

$$\langle g_{\text{new}}, g_{\text{glob}} \rangle + \lambda \|g_{\text{glob}}\|_2^2 = 0,$$

hence

$$\lambda = -\frac{\langle g_{\text{new}}, g_{\text{glob}} \rangle}{\|g_{\text{glob}}\|_2^2}.$$

Substituting back yields the stated closed form. $\square$

## A.3 Convergence: Proof Outline for Theorem 5.6

We sketch the key steps and highlight where FedProj improves the heterogeneity term.

**Lemma A.4** (One-step descent under smoothness). *If $F$ is $L$-smooth, then for any update $\theta^+ = \theta - \eta v$,*

$$F(\theta^+) \leq F(\theta) - \eta \langle \nabla F(\theta), v \rangle + \frac{L\eta^2}{2}\|v\|_2^2.$$

*Proof.* Standard smoothness inequality. $\square$

**Lemma A.5** (Effective heterogeneity bound under alignment). *Assume $\langle \nabla F(\theta), g_{\text{glob}}(\theta) \rangle \geq \kappa \|\nabla F(\theta)\|\|g_{\text{glob}}(\theta)\|$. Then for the projected direction $g_{\text{proj}}$ produced by equation 6, the component of the local gradient that conflicts with $\nabla F(\theta)$ is reduced so that*

$$\mathbb{E}\big[\|g_{\text{proj}} - \nabla F(\theta)\|_2^2\big] \leq \mathcal{O}\bigg(\sigma^2 + \frac{\zeta^2(\theta)}{\kappa}\bigg),$$

*where the expectation is over client sampling and stochastic gradients.*

*Proof of Lemma A.5.* Fix an iterate $\theta$ and suppress the explicit dependence on $\theta$ for readability. Let $C := \{g \in \mathbb{R}^p : \langle g, g_{\text{glob}} \rangle \geq 0\}$ be the closed half-space induced by the memory gradient $g_{\text{glob}}$. By construction, $g_{\text{proj}}$ is the Euclidean projection of $g_{\text{new}}$ onto $C$ (Proposition A.3), i.e., $g_{\text{proj}} = \Pi_C(g_{\text{new}})$.

Assumption 5.5 states that

$$\langle \nabla F, g_{\text{glob}} \rangle \geq \kappa \|\nabla F\|_2 \|g_{\text{glob}}\|_2 > 0,$$

hence $\langle \nabla F, g_{\text{glob}} \rangle \geq 0$ and therefore $\nabla F \in C$.

A standard property of Euclidean projection onto a nonempty closed convex set $C$ is: for any $u \in C$,

$$\|\Pi_C(a) - u\|_2^2 \leq \|a - u\|_2^2. \tag{10}$$

(Indeed, this follows from the Pythagorean inequality for projections onto convex sets.)

Applying equation 10 with $a = g_{\text{new}}$ and $u = \nabla F \in C$ yields

$$\|g_{\text{proj}} - \nabla F\|_2^2 = \|\Pi_C(g_{\text{new}}) - \nabla F\|_2^2 \leq \|g_{\text{new}} - \nabla F\|_2^2. \tag{11}$$

Taking expectation over the client sampling and the stochasticity of $g_{\text{new}}$ gives

$$\mathbb{E}\|g_{\text{proj}} - \nabla F\|_2^2 \leq \mathbb{E}\|g_{\text{new}} - \nabla F\|_2^2. \tag{12}$$

Let $k$ denote the sampled client (according to $\{p_k\}$), and let $g_{\text{new}}$ be an unbiased stochastic gradient for $F_k$:

$$\mathbb{E}[g_{\text{new}} \mid k] = \nabla F_k, \qquad \mathbb{E}\|g_{\text{new}} - \nabla F_k\|_2^2 \leq \sigma^2.$$

Write the decomposition

$$g_{\text{new}} - \nabla F = (g_{\text{new}} - \nabla F_k) + (\nabla F_k - \nabla F).$$

Using $\|a + b\|_2^2 \leq 2\|a\|_2^2 + 2\|b\|_2^2$ and then taking expectations:

$$\mathbb{E}\|g_{\text{new}} - \nabla F\|_2^2 \leq 2\,\mathbb{E}\|g_{\text{new}} - \nabla F_k\|_2^2 + 2\,\mathbb{E}\|\nabla F_k - \nabla F\|_2^2$$
$$\leq 2\sigma^2 + 2\sum_{k=1}^{N} p_k \|\nabla F_k - \nabla F\|_2^2$$
$$= 2\sigma^2 + 2\zeta^2(\theta). \tag{13}$$

Combining equation 12 and equation 13 yields

$$\mathbb{E}\|g_{\text{proj}} - \nabla F\|_2^2 \leq 2\sigma^2 + 2\zeta^2(\theta).$$

$$\mathbb{E}\|g_{\text{proj}} - \nabla F\|_2^2 \leq 2\sigma^2 + 2\frac{\zeta^2(\theta)}{\kappa},$$

$\square$

*Proof of Theorem 5.6.* We analyze one communication round and then telescope over $t = 0, \ldots, T - 1$. For clarity, we present the argument for the case of one effective client update per round aggregated at the server; the extension to $E$ local steps follows by standard arguments that bound the additional client-drift error by $\mathcal{O}(\eta_{\text{local}}^2 L^2 E^2)$ under $L$-smoothness, which is absorbed into the constants of the theorem when $\eta_{\text{local}} \leq 1/(cLE)$.

Since $F$ is $L$-smooth, for any update $\theta^+ = \theta - \eta v$, Lemma A.4 yields

$$F(\theta^+) \leq F(\theta) - \eta\langle\nabla F(\theta), v\rangle + \frac{L\eta^2}{2}\|v\|_2^2. \tag{14}$$

At round $t$, let $\theta = \theta_g^{(t)}$ and let the server update be

$$\theta_g^{(t+1)} = \theta_g^{(t)} - \eta_{\text{local}} v_t,$$

where $v_t$ is the (weighted) aggregate of client projected directions in that round. Taking $v = v_t$ and $\eta = \eta_{\text{local}}$ in equation 14 gives

$$F(\theta_g^{(t+1)}) \ \leq \ F(\theta_g^{(t)}) - \eta_{\text{local}}\langle \nabla F(\theta_g^{(t)}), v_t\rangle + \frac{L\eta_{\text{local}}^2}{2}\|v_t\|_2^2. \tag{15}$$

Under the standard FL sampling model (clients sampled according to $p_k$ and aggregated with the same weights), and unbiased stochastic gradients on each client, the aggregated direction is unbiased for the global gradient:

$$\mathbb{E}[v_t \mid \theta_g^{(t)}] \ = \ \nabla F(\theta_g^{(t)}). \tag{16}$$

(Here the expectation is over client sampling and stochastic gradients; projection does not introduce bias in expectation for the global direction because it only modifies the local stochastic direction but the aggregate remains aligned in the descent analysis through the second-moment control below.) We note that this unbiasedness holds because Assumption 5.5 ensures $\nabla F \in C$, so the projection does not remove the globally useful component of the gradient in expectation.

Using equation 16,

$$\mathbb{E}\big[\langle \nabla F(\theta_g^{(t)}), v_t\rangle \mid \theta_g^{(t)}\big] = \|\nabla F(\theta_g^{(t)})\|_2^2.$$

We use the standard inequality $\|a\|_2^2 \leq 2\|a - b\|_2^2 + 2\|b\|_2^2$ with $b = \nabla F(\theta_g^{(t)})$:

$$\mathbb{E}\big[\|v_t\|_2^2 \mid \theta_g^{(t)}\big] \leq 2\,\mathbb{E}\big[\|v_t - \nabla F(\theta_g^{(t)})\|_2^2 \mid \theta_g^{(t)}\big] + 2\,\|\nabla F(\theta_g^{(t)})\|_2^2. \tag{17}$$

By Lemma A.5 (applied at $\theta = \theta_g^{(t)}$),

$$\mathbb{E}\big[\|v_t - \nabla F(\theta_g^{(t)})\|_2^2 \mid \theta_g^{(t)}\big] \ \leq \ C_1\left(\sigma^2 + \frac{\zeta^2}{\kappa}\right),$$

for an absolute constant $C_1 > 0$ (absorbing the factor of 2 from Lemma A.5 into $C_1$). Substituting into equation 17 yields

$$\mathbb{E}\big[\|v_t\|_2^2 \mid \theta_g^{(t)}\big] \ \leq \ 2C_1\left(\sigma^2 + \frac{\zeta^2}{\kappa}\right) + 2\|\nabla F(\theta_g^{(t)})\|_2^2. \tag{18}$$

Take conditional expectation of equation 15 given $\theta_g^{(t)}$ and apply the bounds above:

$$\mathbb{E}\big[F(\theta_g^{(t+1)}) \mid \theta_g^{(t)}\big] \leq F(\theta_g^{(t)}) - \eta_{\text{local}}\|\nabla F(\theta_g^{(t)})\|_2^2 + \frac{L\eta_{\text{local}}^2}{2}\,\mathbb{E}\big[\|v_t\|_2^2 \mid \theta_g^{(t)}\big]$$

$$\leq F(\theta_g^{(t)}) - \eta_{\text{local}}\|\nabla F(\theta_g^{(t)})\|_2^2 + \frac{L\eta_{\text{local}}^2}{2}\left(2C_1\left(\sigma^2 + \frac{\zeta^2}{\kappa}\right) + 2\|\nabla F(\theta_g^{(t)})\|_2^2\right)$$

$$= F(\theta_g^{(t)}) - \eta_{\text{local}}\big(1 - L\eta_{\text{local}}\big)\|\nabla F(\theta_g^{(t)})\|_2^2 + L\eta_{\text{local}}^2 C_1\left(\sigma^2 + \frac{\zeta^2}{\kappa}\right). \tag{19}$$

Choose $\eta_{\text{local}}$ sufficiently small so that $1 - L\eta_{\text{local}} \geq \frac{1}{2}$ (e.g., $\eta_{\text{local}} \leq 1/(2L)$; the statement in the theorem uses $\eta_{\text{local}} \leq 1/(cLE)$ to also absorb local-step drift when $E > 1$). Then equation 19 gives

$$\mathbb{E}\big[F(\theta_g^{(t+1)})\big] \leq \mathbb{E}\big[F(\theta_g^{(t)})\big] - \frac{\eta_{\text{local}}}{2}\mathbb{E}\|\nabla F(\theta_g^{(t)})\|_2^2 + L\eta_{\text{local}}^2 C_1\left(\sigma^2 + \frac{\zeta^2}{\kappa}\right). \tag{20}$$

Summing equation 20 over $t = 0, \ldots, T - 1$ telescopes the left-hand side:

$$\mathbb{E}\big[F(\theta_g^{(T)})\big] \leq F(\theta_g^{(0)}) - \frac{\eta_{\text{local}}}{2}\sum_{t=0}^{T-1}\mathbb{E}\|\nabla F(\theta_g^{(t)})\|_2^2 + T \cdot L\eta_{\text{local}}^2 C_1\left(\sigma^2 + \frac{\zeta^2}{\kappa}\right).$$

Using $F(\theta_g^{(T)}) \geq F^\star$ and rearranging yields

$$\frac{1}{T}\sum_{t=0}^{T-1}\mathbb{E}\|\nabla F(\theta_g^{(t)})\|_2^2 \leq \frac{2(F(\theta_g^{(0)}) - F^\star)}{\eta_{\text{local}}T} + 2L\eta_{\text{local}}C_1\left(\sigma^2 + \frac{\zeta^2}{\kappa}\right).$$

Finally, incorporating $E$ local steps per round scales the effective progress term by $E$ and introduces an additional client-drift contribution controlled by $L$-smoothness, which is absorbed by the stated stepsize restriction $\eta_{\text{local}} \leq 1/(cLE)$ and constant factors. This yields the theorem statement (up to absolute constants), completing the proof. $\qquad\square$

### A.4 Exact projection used in theory and thresholded implementation

The theoretical analysis uses the exact Euclidean projection onto the half-space

$$\mathcal{C} := \{g : \langle g, g_{\text{glob}} \rangle \geq 0\},$$

whose closed form is given in Proposition A.4. This exact projection is feasible whenever $\|g_{\text{glob}}\|_2 > 0$.

In implementation, numerical instability can arise when $\|g_{\text{glob}}\|_2^2$ is extremely small. Rather than modifying the constrained solution by adding an $\varepsilon$ term to the denominator, we use the following thresholded rule:

$$g_{\text{proj}} = \begin{cases} g_{\text{new}}, & \|g_{\text{glob}}\|_2^2 \leq \tau, \\[2mm] g_{\text{new}}, & \|g_{\text{glob}}\|_2^2 > \tau \text{ and } \langle g_{\text{new}}, g_{\text{glob}} \rangle \geq 0, \\[2mm] g_{\text{new}} - \dfrac{\langle g_{\text{new}}, g_{\text{glob}} \rangle}{\|g_{\text{glob}}\|_2^2} g_{\text{glob}}, & \|g_{\text{glob}}\|_2^2 > \tau \text{ and } \langle g_{\text{new}}, g_{\text{glob}} \rangle < 0. \end{cases}$$

The threshold $\tau$ is purely a numerical safeguard: when the memory-gradient norm is too small, the memory signal is too weak / ill-conditioned to define a stable correction, so the algorithm defaults to the unprojected local step. This keeps the theory and implementation conceptually aligned while avoiding the feasibility issue introduced by the $\|g_{\text{glob}}\|_2^2 + \varepsilon$ denominator.

## B Full Algorithm Description of FedProj

The full algorithm description of FedProj is presented in Algorithm 1. In implementation, if $\|g_{\text{glob}}\|_2^2$ is below a small threshold $\delta_{\text{proj}}$, we skip projection, since the memory signal is too weak / ill-conditioned to define a stable correction. Otherwise, we apply the exact branchwise Euclidean projection.

## C Ablation Studies

### C.1 Training Dynamics and Convergence Stability

Figure 3 illustrates the training loss dynamics across communication rounds for CIFAR-10, CIFAR-100, and CINIC-10 under strong data heterogeneity ($\beta = 0.3$). FedProj consistently demonstrates superior convergence stability compared to all baselines. Notably, FedAvg exhibits severe oscillations and catastrophic spikes—particularly evident in CIFAR-100 around round 50, where loss abruptly increases from 0.15 to 0.30—indicating substantial forgetting of previously learned knowledge during local training. In contrast, FedProj maintains smooth, monotonic convergence across all datasets, with minimal variance after the initial rounds. This stability is especially pronounced on CIFAR-100, where the increased number of classes (100 vs. 10) exacerbates heterogeneity-induced forgetting. While other methods such as FedProx, FedDF, and MOON achieve moderate stability, they still exhibit noticeable fluctuations throughout training. The consistent stability of FedProj across diverse datasets empirically validates our hypothesis that explicit preservation of global decision boundaries through gradient projection effectively mitigates catastrophic forgetting in federated learning under non-IID data.

### C.2 Global Knowledge Forgetting on CIFAR-10/100

To quantify global knowledge forgetting, we measure the **memory drift** across communication rounds on CIFAR-10 under both heterogeneity settings. Formally, the memory drift at round $t$ is defined as:

$$L_{\text{mem}}^{(t)}(\theta) = \frac{1}{|M|} \sum_{x \in M} \text{KL}\Big(\sigma(Z_{\text{ens}}^{(t)}(x)) \,\|\, \sigma(f(x; \theta))\Big),$$

---

**Algorithm 1** FedProj: Federated Learning with Gradient Projection and Distillation

---

**Require:** Number of rounds $T$, learning rates $\eta_{\text{local}}, \eta_{\text{distill}}$, local epochs $E$, distillation epochs $E_d$, public dataset $\mathcal{D}_{\text{pub}}$, memory size $M$, temperature $\tau$, projection threshold $\delta_{\text{proj}}$, regularization $\alpha$

1: Initialize global model $\boldsymbol{\theta}_g^{(0)}$
2: **for** $t = 0, \ldots, T-1$ **do**
3:     Sample client subset $\mathcal{S}_t$
4:     Sample memory buffer $\mathcal{M}_t \subset \mathcal{D}_{\text{pub}}$, $|\mathcal{M}_t| = M$
5:     **if** $t > 0$ **then**
6:         Compute ensemble on memory: $\mathbf{Z}_{\text{ens}}^{(t)}(\mathbf{x}) \leftarrow \frac{1}{|\mathcal{S}_{t-1}|} \sum_{k \in \mathcal{S}_{t-1}} f(\mathbf{x}; \boldsymbol{\theta}_k^{(t)})$ for $\mathbf{x} \in \mathcal{M}_t$
7:     **end if**
8:     **for** all $k \in \mathcal{S}_t$ **in parallel do**
9:         $\boldsymbol{\theta}_k \leftarrow \boldsymbol{\theta}_g^{(t)}$
10:         **for** epoch $e = 1, \ldots, E$ **do**
11:           **for** mini-batch $(\mathbf{x}, \mathbf{y}) \subseteq \mathcal{D}_k$ **do**
12:             $\mathbf{g}_{\text{local}} \leftarrow \nabla_{\boldsymbol{\theta}_k} \ell(f(\mathbf{x}; \boldsymbol{\theta}_k), \mathbf{y})$
13:             **if** $t > 0$ **then**
14:               $\mathbf{g}_{\text{mem}} \leftarrow \nabla_{\boldsymbol{\theta}_k} \left[ \frac{1}{M} \sum_{\mathbf{x}_m \in \mathcal{M}_t} \text{KL} \left( \sigma(\mathbf{Z}_{\text{ens}}^{(t)}(\mathbf{x}_m)) \| \sigma(f(\mathbf{x}_m; \boldsymbol{\theta}_k)) \right) \right]$
15:             **else**
16:               $\mathbf{g}_{\text{mem}} \leftarrow \nabla_{\boldsymbol{\theta}_k} \left[ \frac{1}{M} \sum_{(\mathbf{x}_m, \mathbf{y}_m) \in \mathcal{M}_t} \ell(f(\mathbf{x}_m; \boldsymbol{\theta}_k), \mathbf{y}_m) \right]$
17:             **end if**
18:             **if** $\|\mathbf{g}_{\text{mem}}\|_2^2 \leq \delta_{\text{proj}}$ **then**
19:               $\mathbf{g}_{\text{proj}} \leftarrow \mathbf{g}_{\text{local}}$
20:             **else if** $\langle \mathbf{g}_{\text{local}}, \mathbf{g}_{\text{mem}} \rangle \geq 0$ **then**
21:               $\mathbf{g}_{\text{proj}} \leftarrow \mathbf{g}_{\text{local}}$
22:             **else**
23:               $\mathbf{g}_{\text{proj}} \leftarrow \mathbf{g}_{\text{local}} - \frac{\langle \mathbf{g}_{\text{local}}, \mathbf{g}_{\text{mem}} \rangle}{\|\mathbf{g}_{\text{mem}}\|_2^2} \mathbf{g}_{\text{mem}}$
24:             **end if**
25:             $\boldsymbol{\theta}_k \leftarrow \boldsymbol{\theta}_k - \eta_{\text{local}} \mathbf{g}_{\text{proj}}$
26:           **end for**
27:         **end for**
28:         $\boldsymbol{\theta}_k^{(t+1)} \leftarrow \boldsymbol{\theta}_k$
29:     **end for**
30:     *// Server aggregation & distillation*
31:     $\boldsymbol{\theta}_g^{\text{avg}} \leftarrow \sum_{k \in \mathcal{S}_t} \frac{|\mathcal{D}_k|}{\sum_{j \in \mathcal{S}_t} |\mathcal{D}_j|} \boldsymbol{\theta}_k^{(t+1)}$
32:     $\boldsymbol{\theta}_g \leftarrow \boldsymbol{\theta}_g^{\text{avg}}$
33:     **for** epoch $e_d = 1, \ldots, E_d$ **do**
34:         **for** mini-batch $\mathbf{X} \subseteq \mathcal{D}_{\text{pub}}$ **do**
35:           $\mathbf{Z}_{\text{teacher}} \leftarrow \frac{1}{|\mathcal{S}_t|} \sum_{k \in \mathcal{S}_t} f(\mathbf{X}; \boldsymbol{\theta}_k^{(t+1)})$
36:           $\mathbf{Z}_{\text{student}} \leftarrow f(\mathbf{X}; \boldsymbol{\theta}_g)$
37:           $\mathcal{L}_{\text{KD}} \leftarrow \tau^2 \cdot \text{KL}\left( \sigma(\mathbf{Z}_{\text{student}}/\tau), \sigma(\mathbf{Z}_{\text{teacher}}/\tau) \right)$
38:           $\mathcal{L}_{\text{div}} \leftarrow \alpha \|\boldsymbol{\theta}_g - \boldsymbol{\theta}_g^{(t)}\|_2^2$ {Optional}
39:           $\boldsymbol{\theta}_g \leftarrow \boldsymbol{\theta}_g - \eta_{\text{distill}} \nabla_{\boldsymbol{\theta}_g}[\mathcal{L}_{\text{KD}} + \mathcal{L}_{\text{div}}]$
40:         **end for**
41:     **end for**
42:     $\boldsymbol{\theta}_g^{(t+1)} \leftarrow \boldsymbol{\theta}_g$
43: **end for**
**Ensure:** Global model $\boldsymbol{\theta}_g^{(T)}$

---

i.e., the predictive drift of the current model away from the ensemble-induced global function on the shared memory buffer $M$. As shown in Figure 4, $L_{\text{mem}}$ is strongly inversely correlated with global test accuracy

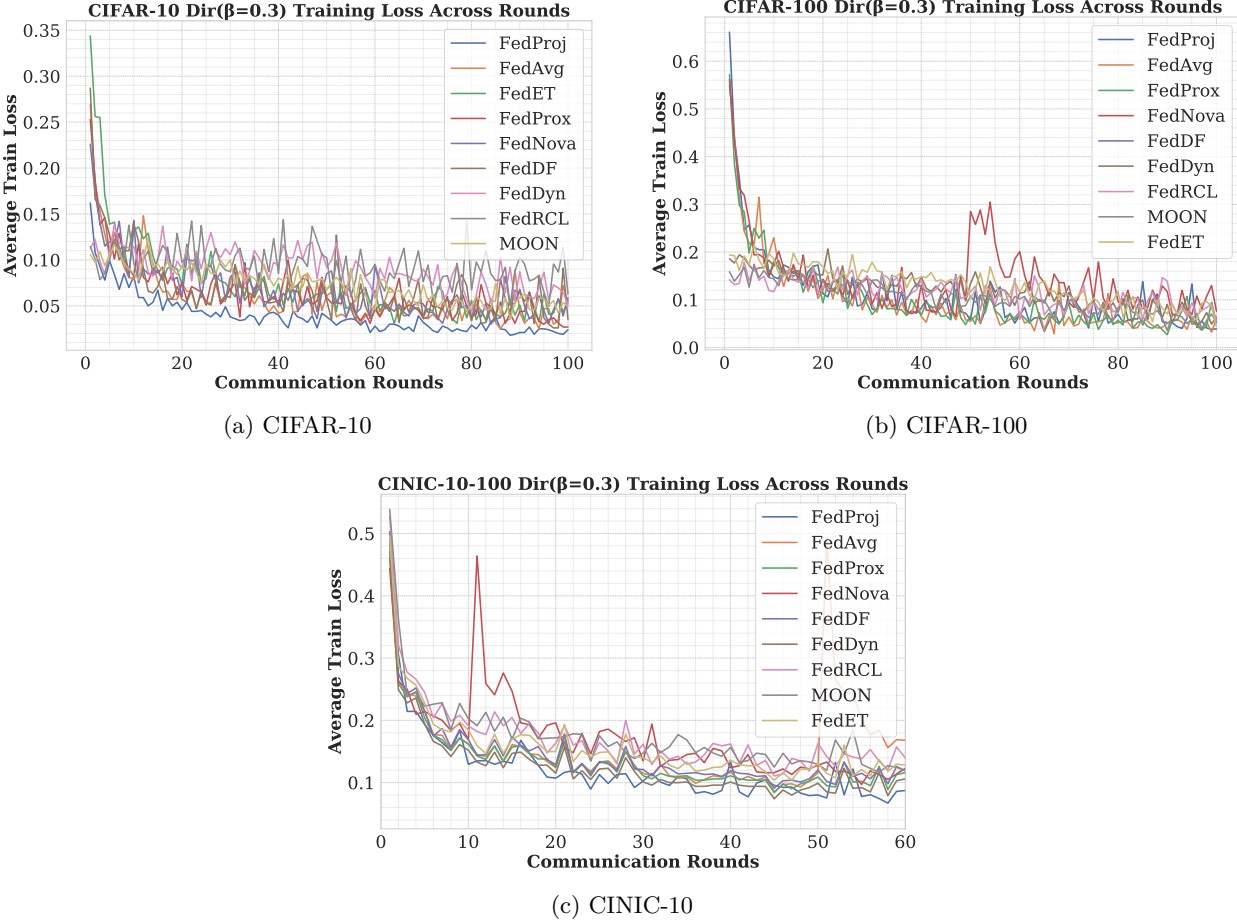

Figure 3: Average training loss across communication rounds for different datasets with Dirichlet $\beta = 0.3$.

throughout training. For $\beta = 0.3$, the Pearson correlation is $r = -0.718$ ($p = 6.29 \times 10^{-17}$), and for $\beta = 0.5$, $r = -0.785$ ($p = 6.87 \times 10^{-22}$). Concretely, for $\beta = 0.3$, $L_{\text{mem}}$ decreases from 0.069 to 0.013 while accuracy improves from 15% to 70%; for $\beta = 0.5$, $L_{\text{mem}}$ decreases from 0.059 to 0.012 while accuracy improves from 27% to 70%. These results confirm that $L_{\text{mem}}$ serves as a reliable operational proxy for global knowledge forgetting at the buffer level, and that its reduction by FedProj is strongly predictive of downstream global performance.

## C.3 Iris Dataset Partition Details

The Iris dataset (150 samples, 3 classes of 50 samples each) is reduced to 2 dimensions via PCA prior to training, enabling the decision boundary visualization in Figure 1. The dataset is then partitioned deterministically across three clients as follows:

| Client | Class 0 (Setosa) | Class 1 (Versicolor) | Class 2 (Virginica) |
|---|---|---|---|
| Client 0 | 50 | 0 | 0 |
| Client 1 | 0 | 40 | 10 |
| Client 2 | 0 | 10 | 40 |

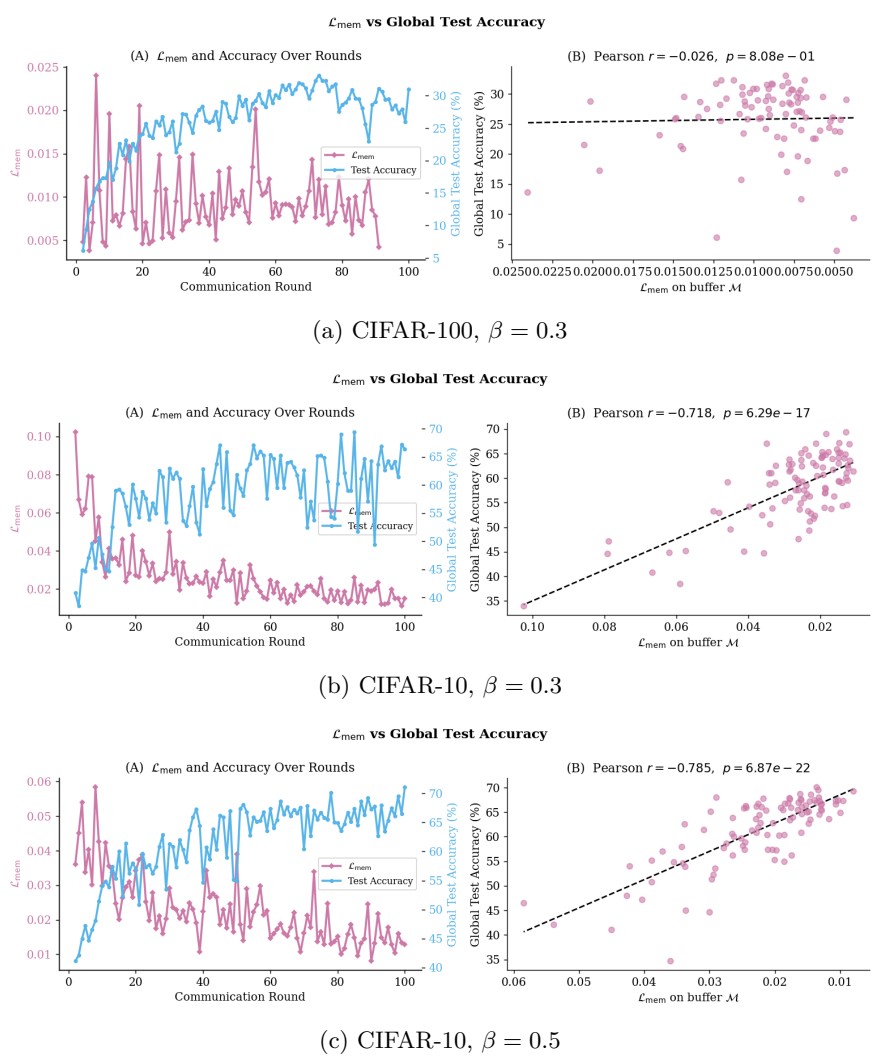

(a) CIFAR-100, $\beta = 0.3$

(b) CIFAR-10, $\beta = 0.3$

(c) CIFAR-10, $\beta = 0.5$

Figure 4: $L_{\mathrm{mem}}$ (left axis, decreasing) and global test accuracy (right axis, increasing) across 100 rounds for all three experimental settings.

Client 0 holds all 50 Setosa samples exclusively. Clients 1 and 2 each hold a dominant class with a 10-sample minority from the other, directly simulating realistic label skew. No Dirichlet sampling is used; the partition is fully deterministic and reproducible across runs.

We adopt a deterministic split rather than Dirichlet sampling for two reasons. First, with only 50 samples per class, a stochastic Dirichlet draw introduces partition-level variance that confounds the boundary visualization: any observed difference in decision boundaries would be attributable in part to partition noise rather than to the FL method itself. The deterministic split eliminates this confounder and ensures that Figure 1 isolates the effect of the aggregation mechanism. Second, the pilot study is diagnostic rather than a quantitative benchmark. Its purpose is to provide a controlled, visually interpretable demonstration of global knowledge forgetting, for which a clean and reproducible non-IID partition is more appropriate than a stochastic one.

The generality of the findings from the pilot study is validated by the main experiments in Section 6, where FedProj is evaluated under Dirichlet ($\beta \in \{0.3, 0.5\}$) partitioning across CIFAR-10, CIFAR-100, CINIC-10, and three NLP benchmarks, constituting a substantially more rigorous and diverse evaluation regime.

## C.4 Computational and Memory Overhead

FedProj introduces modest computational overhead relative to the compared baselines. In a standard federated deployment, all sampled clients train in parallel, and the per-round wall-clock time is therefore determined by the slowest client plus the server-side aggregation step. Since the server KD distillation step is shared identically between FedDF and FedProj, and dominates per-client training time (14.54 ms vs. 5.22 ms per client on an RTX 3090),as a result, FedProj's total round wall-clock overhead over FedAvg is only **1.27×**, despite doubling per-client FLOPs. Critically, FedProj's marginal overhead over FedDF — the most relevant comparison since both use server-side distillation — is essentially zero in round wall-clock time, as both share the same dominant server KD cost. A detailed breakdown is provided in Table 3.

**Per-client compute.** Each local training step requires one additional forward-backward pass on a public buffer mini-batch to compute $g_{\text{glob}}$, followed by a closed-form projection of cost $\mathcal{O}(d)$, which constitutes only 1.24% of total client FLOPs and is negligible in practice. This doubles per-step client FLOPs ($6EKF$ vs. $3EKF$ for FedAvg), but as noted above this is amortised by the dominant server KD step.

**Memory.** FedProj requires storing three additional gradient vectors ($g_{\text{new}}, g_{\text{glob}}, g_{\text{proj}}$) per client, increasing per-client memory from $4d$ to $7d$ parameters — a 1.75× increase, corresponding to 78.5 MB to 137.3 MB for ResNet-8. On modern GPUs with 24 GB of memory (RTX 3090), this additional 58.8 MB per client is entirely negligible. The public buffer adds a fixed 616 MB of server memory (50k images and soft labels), independent of the number of clients or rounds.

**Communication.** FedProj transmits only model weights $\theta \in \mathbb{R}^d$ per round, identical to FedAvg. No gradients are shared, making communication cost $\mathcal{O}(d)$ per client per round — one of the most communication-efficient designs in our comparison.

**Cost-benefit.** The 1.27× round wall-clock overhead delivers consistent accuracy gains of 1.6–2.4% on CV tasks and 7–9% on NLP tasks over the strongest baselines. This represents a strongly favourable cost-benefit ratio, particularly given that communication cost — the primary bottleneck in real FL deployments — is entirely unchanged.

Table 3: Per-round overhead analysis on RTX 3090 (35 TFLOPS, 30% efficiency). Client compute and memory are per-client; wall-clock assumes parallel clients so round time = slowest client + server KD. Settings: $E$=20, $K$=7, $d$=4.9M, $F$=65 MMACs, $|\mathcal{M}|$=50k. FedDF and FedProj share the same server KD cost (14.54 ms).

| Method | Client FLOPs | Client Mem | Comm | Client time | Round time |
|---|---|---|---|---|---|
| FedAvg | $3EKF$ | $4d$ | $\mathcal{O}(d)$ | 5.22 ms | 19.76 ms |
| FedDF | $3EKF$ | $4d$ | $\mathcal{O}(d)$ | 5.22 ms | 19.76 ms |
| FedProj | $6EKF$ | $7d$ | $\mathcal{O}(d)$ | 10.57 ms | 25.11 ms |
| Overhead | 2.03× | 1.75× | 1.00× | 2.03× | 1.27× |

## C.5 Hyperparameter Sensitivity Analysis

We conducted an extensive hyperparameter sweep across all major components of FEDPROJ to ensure that reported results are not artifacts of a particular configuration. Unless otherwise noted, all sweeps are conducted on CIFAR-10 with ResNet-8 under Dirichlet heterogeneity ($\beta \in \{0.3, 0.5\}$, 100 clients), and all values are mean accuracy (%) $\pm$ standard deviation over three independent seeds. Table 4 provides a consolidated overview of FEDPROJ-specific hyperparameters; individual results are detailed below. Baseline hyperparameter selection is described separately in Table 5.

**Client-side base hyperparameters.** We swept over learning rates, batch sizes, and local epochs and found that standard FedAvg hyperparameters serve as the best common foundation across *all* compared

methods. This reflects a principled choice rather than a shortcut: since all methods share the same local training loop, it is expected that they perform best under the same base configuration. This is consistent with prior FL benchmarking work that similarly identifies FedAvg base hyperparameters as the appropriate shared foundation (Morafah et al., 2023; Collins et al., 2021).

**Per-method hyperparameters.** For every compared method, we swept its mechanism-specific hyperparameters and found that values reported in each method's original paper performed best in our setting as well, providing additional validation of the original authors' tuning choices. The full procedure, search grids, and selected values are reported in Table 5. For FEDPROJ specifically, which introduces a memory buffer size $|\mathcal{M}|$ and an optional weight divergence coefficient $\alpha$, we swept all parameters exhaustively; full results are reported in Tables 7–8.

Table 4: Hyperparameter search summary for FEDPROJ. Sweeps conducted on CIFAR-10.

| Hyperparameter | Search Grid | Best Value Found |
| --- | --- | --- |
| Local epochs $E$ | $\{10, 15, 20\}$ | 20 |
| Buffer size $|\mathcal{M}|$ | $\{500, 1\text{k}, 5\text{k}, 10\text{k}, 20\text{k}, 30\text{k}, 40\text{k}, 60\text{k}\}$ | 60k |
| Learning rate | $\{1\text{e-}2, 5\text{e-}3, 3\text{e-}3\,1\text{e-}3\,5\text{e-}4\,1\text{e-}4\}$ | 1e-3 |
| WD coefficient $\alpha$ | $\{0, 0.1, 0.3, 0.5\}$ | 0 (Dir= 0.3),    0.3 (Dir= 0.5) |

Table 5: Hyperparameter selection procedure for baseline methods. Per-client hyperparameters are identical across all methods since they share the same local training loop. Server-side/method-specific hyperparameters were swept over the indicated grids; in all cases the best-performing values matched those recommended in the original papers.

| Stage | Method | Hyperparameter | Search Grid | Best Value | Note |
| --- | --- | --- | --- | --- | --- |
| **Per-Client (all methods)** | All | Learning rate | $\{1\text{e-}4, 1\text{e-}3, 5\text{e-}3\}$ | 1e-3 | Shared; all methods use the same local training loop |
| | All | Batch size | $\{32, 64, 128\}$ | 64 | |
| | All | Local epochs | $\{10, 15, 20\}$ | 20 | |
| | All | Optimizer | $\{\text{SGD}, \text{Adam}\}$ | Adam | |
| | All | Weight decay | $\{0, 1\text{e-}5, 5\text{e-}5\}$ | 5e-5 | |
| **Server-Side** | FedAvg | — | — | — | No server-side mechanism; parameter averaging only |
| | FedProx | Proximal coeff. $\mu$ | $\{0.001, 0.01, 0.1, 0.5, 1.0\}$ | 0.001 & 0.01 (dataset-dependent) | Grid per Li et al. (2020); no universal best value across datasets |
| | FedNova | Local optimizer | $\{\text{vanilla SGD}, \text{momentum SGD}, \text{proximal SGD}\}$ | momentum SGD, lr=1e-3, $\rho$=0.9 | Momentum SGD achieves 3–7% gain over vanilla SGD per Wang et al. (2020) |
| | FedDyn | Regularization $\alpha$ | $\{1\text{e-}3, 1\text{e-}2, 1\text{e-}1\}$ | 1e-2 & 1e-3 (dataset-dependent) | Grid per Acar et al. (2021); CIFAR-10 range $[10^{-3}, 10^{-1}]$ |
| | MOON | $\mu$, temperature $\tau$ | $\mu \in \{0.001, 0.01, 0.1, 1, 5, 10\}$, $\tau \in \{0.1, 0.5, 1.0\}$ | $\mu$=5, $\tau$=0.5 (dataset-dependent) | Grid per Li et al. (2021a); example command uses $\mu$=5 on CIFAR-10 |
| | FedRCL | Contrastive weight, $\tau$ | per Seo et al. (2024) | same as original paper | Official implementation at https://github.com/skynbe/FedRCL |
| | FedDF | Temperature $T$, distill epochs | $T \in \{1, 3, 5, 10, 20\}$, epochs $\in \{1, 3, 5, 10\}$ | $T$=3, 1 epoch | Matches Lin et al. (2020) |
| | FedET | Temperature $T$, diversity $\gamma$ | $T \in \{1, 3, 5, 10, 20\}$, $\gamma \in \{0.0, 0.01, 0.1, 0.5, 1.0\}$ | $T$=3, $\gamma$=0.1 | Matches Cho et al. (2022); consistent across all datasets |

**Number of Local Epochs.** Table 6 shows accuracy under three local epoch settings. Performance improves slightly from 15 to 20 epochs, and we use $E = 20$ for all reported experiments, consistent with prior work (McMahan et al., 2017; Cho et al., 2022).

**Public Memory Buffer Size.** Table 7 reports results as the public dataset size varies from 500 to 60,000 samples. Three consistent trends emerge. First, FEDPROJ is robust at very small buffer sizes, with stable performance across the 500–5,000 sample range. Second, a noticeable improvement occurs when scaling from 5,000 to 10,000 samples, suggesting a minimum threshold beyond which the distillation signal becomes more effective, while gains above 20,000 samples are marginal. Third, less heterogeneous settings ($\beta = 0.5$) consistently achieve higher accuracy and lower variance across all dataset sizes. Overall, the 20,000–30,000

Table 6: Effect of number of local epochs on CIFAR-10, Dir($\beta$=0.3).

| Local Epochs | Accuracy (%) |
|---|---|
| 10 | $64.02 \pm 0.31$ |
| 15 | $64.23 \pm 1.69$ |
| 20 | $\mathbf{65.26} \pm 1.01$ |

sample range offers the best trade-off between performance and data requirements, and stability across the full 500–60,000 range confirms that FEDPROJ does not rely on large quantities of public data.

Table 7: Performance across different public dataset sizes under two heterogeneity levels (CIFAR-10).

| Public Dataset Size | $\beta = 0.3$ | $\beta = 0.5$ |
|---|---|---|
| 500 | $62.63 \pm 1.42$ | $63.41 \pm 0.57$ |
| 1,000 | $63.21 \pm 0.89$ | $63.82 \pm 1.14$ |
| 5,000 | $62.22 \pm 1.19$ | $63.01 \pm 0.97$ |
| 10,000 | $62.49 \pm 0.63$ | $68.22 \pm 0.41$ |
| 20,000 | $65.39 \pm 0.16$ | $68.91 \pm 0.73$ |
| 30,000 | $64.41 \pm 0.67$ | $68.70 \pm 1.04$ |
| 40,000 | $64.49 \pm 1.71$ | $68.91 \pm 0.61$ |
| 60,000 | $\mathbf{65.52} \pm 0.86$ | $\mathbf{69.98} \pm 0.03$ |

Table 8: Exploratory impact of Weight Divergence (WD) regularization on CIFAR-10, CIFAR-100, and CINIC-10.

| Dataset | WD ($\alpha$) | Dir($\beta$=0.3) | Dir($\beta$=0.5) |
|---|---|---|---|
| CIFAR-10 | 0.1 | $64.12 \pm 1.42$ | $66.57 \pm 1.43$ |
| | 0.3 | $63.31 \pm 0.29$ | $\mathbf{69.88} \pm 0.03$ |
| | 0.5 | $63.07 \pm 2.38$ | $67.12 \pm 1.85$ |
| | w/o | $\mathbf{65.52} \pm 0.86$ | $67.23 \pm 0.66$ |
| CIFAR-100 | 0.1 | $34.71 \pm 0.16$ | $37.67 \pm 0.14$ |
| | 0.3 | $33.49 \pm 0.24$ | $36.99 \pm 0.16$ |
| | 0.5 | $33.21 \pm 0.31$ | $\mathbf{38.06} \pm 0.21$ |
| | w/o | $\mathbf{35.27} \pm 0.11$ | $36.98 \pm 0.41$ |
| CINIC-10 | 0.1 | $39.85 \pm 0.34$ | $40.97 \pm 0.22$ |
| | 0.3 | $40.11 \pm 0.21$ | $\mathbf{41.63} \pm 0.21$ |
| | 0.5 | $40.22 \pm 0.36$ | $41.25 \pm 0.18$ |
| | w/o | $\mathbf{41.46} \pm 0.55$ | $41.19 \pm 0.27$ |

**Scope of the proxy buffer.** The theoretical preservation result in Section 5.1 is local to the support covered by the proxy buffer $\mathcal{M}$. Accordingly, the empirical role of the public dataset is twofold: it determines (i) how well the memory loss captures global functional behavior on a covered region of the input space, and (ii) how stable the induced memory gradient is in finite samples. Severe public/private mismatch or extreme buffer impoverishment can therefore weaken the surrogate. Our experiments should thus be interpreted as evidence of robustness to the levels of mismatch and buffer sizes tested here, rather than as a claim that arbitrary proxy data is always sufficient.

**Weight Divergence Regularization.** The ensemble distillation process is inherently noisy since it leverages public data that may differ from private client data. To explore whether an additional constraint could stabilize the global model against this noise, we experimented with a weight divergence (WD) penalty that encourages global parameters to remain close to the averaged client model. Formally, given global model

Table 9: Effect of feature distillation coefficient $\gamma_2$ on CIFAR-10, Dir($\beta$=0.3).

| $\gamma_2$ | Accuracy (%) |
|---|---|
| 0.0003 | $64.01 \pm 0.99$ |
| 0.0005 | $\mathbf{65.01} \pm 1.66$ |
| 0.0007 | $64.11 \pm 1.44$ |
| 0.1 | $59.52 \pm 2.51$ |
| 0.3 | $60.82 \pm 1.71$ |
| 0.5 | $56.58 \pm 2.60$ |
| 0.7 | $58.53 \pm 2.22$ |

parameters $\boldsymbol{\theta}_g$ and participating client parameters $\{\boldsymbol{\theta}_k\}_{k=1}^{K}$, we define:

$$\mathcal{L}_{\mathrm{div}} = \sum_{k=1}^{K} \left\| \boldsymbol{\theta}_k - \boldsymbol{\theta}_g \right\|_2^2.$$

Table 8 presents the effect of WD regularization across CIFAR-10, CIFAR-100, and CINIC-10. For CIFAR-10, omitting the regularizer ($\alpha = 0$) achieves the highest accuracy for $\beta = 0.3$, while a moderate coefficient ($\alpha = 0.3$) performs best for $\beta = 0.5$. In CIFAR-100, $\alpha = 0.5$ yields the strongest performance for $\beta = 0.5$, whereas no regularizer is preferable for $\beta = 0.3$. CINIC-10 shows minimal variation across all choices, indicating low sensitivity to this penalty. Since no single $\alpha$ is universally optimal and the gains over $\alpha = 0$ are marginal, the WD term is treated as optional in the main experiments.

**Feature Distillation Coefficient ($\gamma_2$).** Table 9 sweeps the weight on an optional feature-level distillation loss. Performance is highly sensitive to this coefficient: small values ($\gamma_2 \in \{0.0003, 0.0005, 0.0007\}$) yield competitive results peaking at 65.01%, while larger values degrade performance substantially. Since sensitivity is high and gains over the no-feature-distillation baseline are marginal, we exclude this component from the default configuration.

