# OpenReview forum: "Towards Preventing Global Knowledge Forgetting in Federated Learning with Non-IID Data"
_TMLR — Accepted by TMLR_

### Review · Reviewer_CXDU · 2026-02-26

**Summary Of Contributions:**

This paper points out “global knowledge forgetting” as a problem in federated learning when clients have different data. During local training, clients overfit their biased data and quickly lose the useful global multi-class structure. As a result, server aggregation averages incompatible models instead of building progress. To fix this, the paper introduces FedProj, which (i) uses a client-side projected gradient update to avoid increasing a KL-based “global memory loss” on a small public or proxy buffer, and (ii) applies server-side ensemble distillation on public data to merge client knowledge into a consistent global model. Experiments show better accuracy on several CV and NLP benchmarks, along with a small visualization study that demonstrates how decision boundaries erode and recover.

**Additional Comments:**

N/A

**Audience:**

Yes

**Audience Explanation:**

Some of TMLR’s readers, especially those focused on federated learning, optimization with heterogeneity, and continual or knowledge-preserving training, will likely find this paper interesting.

**Claims And Evidence:**

No

**Claims Explanation:**

1) Theorem 5.6 and related text claim that FedProj “reduces effective heterogeneity” by scaling the heterogeneity term by $1/\kappa$, where $\kappa$ is their alignment parameter. However, since $\kappa$ is assumed to be in (0,1], $1/\kappa$ is always at least 1, which actually increases the heterogeneity term unless $\kappa = 1$. This chalenges the claim that the projection reduces heterogeneity as currently stated.

Specifically, in Lemma A.7 they first derive a bound that look like
$$
\mathbb{E}|g_{\text{proj}}-\nabla F|^2 \le \mathbb{E}|g_{\text{new}}-\nabla F|^2 \le 2\sigma^2 + 2\zeta^2(\theta),
$$
which does not include $\kappa$ at all. Then they use a looser inequality, $\zeta^2(\theta) \le \zeta^2(\theta)/\kappa$, simply because $\kappa \le 1$, which actually make the bound worse. This does not really support the idea of “reduction”, it’s just a relaxation.

Suggested fix: either (a) find a real $\kappa$-dependent improvment factor that decreases with $\kappa$ (like multiplying by $1-\kappa^2$ or something similar, depending on the geometry), or (b) remove or rephrase the “reduced heterogenity” claim and instead present the result as a non-expansiveness or stability argument.

2) Assumption 5.5 says the memory-loss gradient $g_{\text{glob}}=\nabla_\theta L_{\text{mem}}$ should point roughly in the same direction as the true global objective gradient $\nabla F(\theta)$, with their cosine similarity at least $\kappa$. But in practice, $L_{\text{mem}}$ is computed on a public or proxy buffer using ensemble logits from clients, not the true global objective. There’s no strong reason to expect this gradient to consistently align with $\nabla F$, especially when there’s domain shift between public and client data or severe label skew.

This is important because if the memory gradient is misaligned, the projection constraint might block useful local updates. The convergence theory depends heavily on this alignment.

Suggested fix: measure the cosine similarity between $\nabla F$ and $g_{\text{glob}}$ (or a proxy like alignment with the aggrigated gradient) across different rounds and settings. Also, discuss cases where the public data does not match well and causes problems.)


3. They motivated the constraint $L_{\text{mem}}(\theta’) \le L_{\text{mem}}(\theta)$ and then use a first-order condition $\langle g_{\text{proj}}, g_{\text{glob}}\rangle \ge 0$ to enforce non-increase (given $\theta’=\theta-\eta g_{\text{proj}}$). This is fine as a first-order control, but the paper repeatedely phrases it as a strong guarantee.

Two issues:

* It is only a local linearization guarantee; higher-order terms can increase $L_{\text{mem}}$ unless step sizes are small enough (not explicitly tied to Lipschitz constants of $\nabla L_{\text{mem}}$).
* Their closed form uses a denominator $|g_{\text{glob}}|^2+\varepsilon$. With $\varepsilon>0$, the returned direction need not satisfy the boundary condition exactly in the active case; depending on signs/magnitudes this can slightly violate $\langle g_{\text{proj}}, g_{\text{glob}}\rangle \ge 0$. If $\varepsilon$ is “just for stability,” they should explain how feasibility is preserved (e.g., only add $\varepsilon$ when $|g_{\text{glob}}|^2$ is tiny and then explicitly clamp to feasibility).

Suggested fix: soften the claim of a “hard gaurantee,” add a condition on step size to ensure $L_{\text{mem}}$ truly decreases, and make sure the projection implementation enforces feasibilty or prove that the version with $\varepsilon$ still keeps feasibility.

4) Proposition 5.3 shows that if KL shift on $M$ is small and margins on $M$ are large enough, labels on $M$ are preserved. This is mathematically fine, but:

* It only applies to the buffer $M$, not to the client or test distributions.
* The needed “non-degeneracy” condition (for example, $\min_j p_j \ge p > 0$) may be unrealistic for confident classifiers, where many classes have probabilities near zero, especially when the number of classes $C$ is large.

Suggested fix: explicitly analyze sensitivity to buffer size/coverage and include experiments where the public buffer is small, shifted, or imperfect, and measure whether boundary preservation on $M$ correlates with test stability.

**Requested Changes:**

Please see my comments above

---

> ### Author Response · Authors · 2026-04-04
>
> We thank the reviewer for the careful and technically insightful reading of our submission. We found the comments very helpful, especially in pushing us to sharpen the theoretical claims and better separate three different layers of the paper: (i) what is formally proved, (ii) what is only assumed as a sufficient condition, and (iii) what is supported empirically. We also appreciate the reviewer’s positive assessment that the problem and findings are of interest to the TMLR audience.
>
> Following these comments, we have revised the paper in several important ways. Most notably, we now frame Theorem 5.6 as a non-expansiveness / stability result rather than as a reduction in effective heterogeneity; we no longer describe Assumption 5.5 as empirically validated; we soften the memory-loss claim from a hard guarantee to a first-order safeguard; and we revise Proposition 5.3 to make explicit that it is a buffer-level result, while replacing the original non-degeneracy condition with a cleaner probability-gap argument. We also clarify the practical role of proxy coverage, domain shift, and buffer size, and explicitly discuss these as limitations rather than overextending the theorem.
>
> We believe these revisions substantially improve the precision and trustworthiness of the paper, and we respond to each point in detail below.
>
> ---
>
> **Response to Comment 1**
>
> We thank the reviewer for catching this. We agree that our original wording was incorrect: since $\kappa \in (0,1]$, the factor $1/\kappa$ cannot be interpreted as a reduction in heterogeneity. The reviewer is also correct that Lemma A.7 first gives the stronger bound
>
> $\mathbb{E}[||g_{\text{proj}} - \nabla F||_2^2] \le$
>
> $\mathbb{E}[||g_{\text{new}} - \nabla F||_2^2] \le$
>
> $ 2\sigma^2 + 2\zeta^2(\theta)$
>
> and that the later $1/\kappa$ form is only a looser relaxation.
>
> Following the reviewer’s suggestion, we have removed the "reduces effective heterogeneity" language throughout and now frame the result as a *non-expansiveness* statement: under the stated geometric condition, projection does not move the local update farther from the true global gradient than the unprojected update. We keep the $1/\kappa$ form only for consistency with the convergence bound, and explicitly label it as a relaxation rather than an improvement.
>
> We also added a controlled experiment showing that local-memory gradient conflict is substantial under stronger heterogeneity and that FedProj suppresses this conflict in practice. We now present this figure only as evidence that the projection is often active, not as a direct empirical verification of the theorem.
>
> So the revised claim is narrower but correct: FedProj provides a non-expansive gradient guarantee, rather than a $\kappa$-dependent reduction in effective heterogeneity.
>
> ---
>
> **Response to Comment 2**
>
> We thank the reviewer for this important point. We agree that our earlier wording was too strong. Assumption 5.5 is a *theoretical sufficient condition* involving the true global gradient $\nabla F(\theta)$, which is not directly observable in federated learning because it depends on the full private objective across clients. We therefore do not claim to empirically verify this assumption, and we will remove the wording suggesting that proxy cosine measurements "validate" it.
>
> What we can measure in practice is only a server-side proxy direction, not the true $\nabla F(\theta)$. Moreover, in a high-dimensional nonconvex regime, near-zero cosine values are not by themselves strong evidence of meaningful alignment. We therefore agree that proxy alignment should only be treated as a diagnostic, not as support for the theorem.
>
> In the revision, we will present Assumption 5.5 only as a conditional geometric assumption for the current non-expansiveness proof, and we will explicitly discuss failure cases. In particular, if the public/proxy buffer is severely mismatched to the client distribution, or if the buffer is too small or insufficiently diverse, then the memory gradient may become a weak or misleading signal and projection may suppress useful updates.
>
> Empirically, the point we believe is most relevant is narrower: the memory objective is useful in practice. Across the main settings, lower $L_{\mathrm{mem}}$ is associated with better global accuracy, and FedProj remains effective in the tested cross-domain settings. We will present this as evidence of practical robustness to moderate mismatch, rather than as a proof of Assumption 5.5.

---

> ### Author Response · Authors · 2026-04-04
>
> ---
>
> **Response to Comment 3**
>
> We thank the reviewer for this helpful comment. We agree that our earlier wording was too strong, and we have revised the paper accordingly.
>
> First, we now describe the projection as a *first-order safeguard* rather than a hard guarantee. The constraint $\langle g_{\mathrm{proj}}, g_{\mathrm{glob}} \rangle \ge 0 $
> comes from a first-order view of $L_{\mathrm{mem}}$, while higher-order terms still depend on the local step size and the smoothness of $L_{\mathrm{mem}}$. We therefore removed the previous "hard guarantee" language and clarified this point in Section 4.2.
>
> Second, the reviewer is correct that the denominator $\|\|g_{\mathrm{glob}}\|\|_2^2 + \varepsilon$ does not preserve feasibility exactly in the active case. To address this cleanly, we now separate theory from implementation. In the theory, we use the exact Euclidean projection onto the half-space defined by
>
> $g : \langle g, g_{\mathrm{glob}} \rangle \ge 0$
>
> which is exactly feasible whenever $\|\|g_{\mathrm{glob}}\|\|_2 > 0.$
>
> In the implementation, the purpose of $\varepsilon$ was only numerical stability; in the revision, we remove it from the theory and instead state a thresholded practical rule: if $\|\|g_{\mathrm{glob}}\|\|_2^2$ is too small, we skip projection.
>
> So the revised statement is narrower but correct: FedProj imposes a first-order half-space constraint, not an exact per-step monotonicity guarantee for $L_{\mathrm{mem}}$. Empirically, however, we still observe that FedProj substantially reduces $L_{\mathrm{mem}}$ relative to No-Proj, which supports the practical effectiveness of this safeguard.
>
> ---
>
> **Response to Comment 4**
>
> We thank the reviewer for this important observation. We agree with both points.
>
> The reviewer is correct that Proposition 5.3 is only a buffer-level result: it guarantees preservation of the induced classification on the proxy buffer $M$, not on the full client or test distribution. We will revise the paper to state this explicitly and remove any wording that could be read as a test-distribution guarantee.
>
> We also agree that the original non-degeneracy condition is stronger than necessary. Rather than assuming a lower bound such as $\min_j p_j \ge \bar p > 0$, we will replace the proof with a simpler probability-gap argument based on the top-1/top-2 predictive gap. Using Pinsker's inequality,
>
> $||p(x) - q(x)||_1 \le \sqrt{2\,KL(p(x) || q(x))}$,
>
> one obtains a sufficient condition of the form
>
> $KL(p(x) || q(x)) < \Delta_p(x)^2 / 8$
>
> for label stability at $x$, where $\Delta_p(x)$ is the top-1/top-2 probability gap under $p$. This avoids requiring all class probabilities to be bounded away from zero.
>
> More broadly, we agree that extending preservation beyond $M$ requires a coverage / representativeness assumption. We will make this explicit and note the corresponding limitation: if the public buffer is too small, too shifted, or insufficiently diverse, then it becomes a weak surrogate for the target distribution.
>
> Empirically, the claim we can support is narrower: $L_{\mathrm{mem}}$ is a useful surrogate in practice. In the revised results, lower $L_{\mathrm{mem}}$ is associated with better global accuracy, and the added buffer-size ablations / cross-domain settings suggest practical robustness to the levels of mismatch tested. We will present these findings as practical evidence rather than as a theorem-level guarantee beyond $M$.
>
> ---
>
> **Closing remark**
>
> We thank the reviewer again for the careful and constructive feedback. We believe the revisions above directly address the main concerns and substantially strengthen the paper’s theoretical framing, technical precision, and empirical interpretation. If there are any remaining points that would benefit from further clarification, we would be glad to address them.

---

### Review · Reviewer_8ADf · 2026-03-16

**Summary Of Contributions:**

This paper studies federated learning under client-level non-IID data and argues that a key failure mode is catastrophic forgetting of the global decision boundary during local training. The authors' proposed method, FedProj, has two components. First, on each client, it computes a local gradient and projects it onto a half-space defined by a memory-based distillation gradient. Second, on the server, it performs ensemble distillation on a small dataset to consolidate the preserved client knowledge into a single global model. Experimentally, the paper reports consistent gains over several federated baselines on CIFAR-10/100, CINIC-10, and three NLP tasks.

**Additional Comments:**

NA

**Audience:**

Yes

**Audience Explanation:**

The findings of this paper are likely to be of interest to the federated learning community.

**Claims And Evidence:**

Yes

**Claims Explanation:**

1. The motivation and method are clear, and the visualization is helpful.

2. The authors provide sufficient evidence, including theoretical analysis, extensive experiments and ablation studies, to support the effectiveness of their proposed method.

3. This paper is well-written, making it easy to follow.

**Requested Changes:**

I am not familiar with related fields. After reading the paper, I can probably understand the motivation and technical content.

1. Can the authors provide a measurement of global knowledge forgetting on the CIFAR-10?

2. Could the authors provide a more detailed analysis of the method’s computational and memory overhead?

3. Since representation geometry and optimisation behaviour can differ substantially in transformer-based vision models, could the authors evaluate FedProj on a ViT-based backbone?

---

> ### Author Response · Authors · 2026-03-31
> **Response to reviewer 8ADf**
>
> We sincerely thank Reviewer 8ADf for their positive and constructive
> assessment of our work. We are grateful for the reviewer's recognition
> that the motivation and method are "clear," the visualization is
> "helpful," and that the paper provides "sufficient evidence, including
> theoretical analysis, extensive experiments and ablation studies, to
> support the effectiveness of the proposed method." We address the
> three constructive requests below point by point.
>
> ---
>
> **Response to comment 2**
>
> We provide a detailed analysis of FedProj's overhead, summarized in
> Table 1 below. The key takeaway is that FedProj's practical overhead
> is modest.
>
> In a real federated deployment, all sampled clients train in parallel,
> so the round wall-clock time equals the slowest client plus the server
> distillation step. The server KD distillation step — shared identically
> between FedDF and FedProj — dominates client training time (14.54 ms
> vs. 5.22 ms per client on an RTX 3090). As a result, FedProj's total
> round wall-clock overhead over FedAvg is only 1.27×, despite
> doubling per-client FLOPs. Critically, FedProj's marginal overhead
> over FedDF — the most relevant comparison since both use server-side
> distillation — is essentially zero in round wall-clock time, as both
> share the same dominant server KD cost.
>
> **Per-client compute.** Each local training step requires one
> additional forward-backward pass on a public buffer mini-batch to
> compute $g_{\text{glob}}$, followed by a closed-form projection of
> cost $O(d)$ (1.24% of total client FLOPs, negligible). This doubles
> per-step client FLOPs (6EKF vs. 3EKF for FedAvg), but as noted above
> this is amortized by the dominant server KD step in practice.
>
> **Memory.** FedProj requires storing three additional gradient vectors
> ($g_{\text{new}}$, $g_{\text{glob}}$, $g_{\text{proj}}$) per client,
> increasing per-client memory from $4d$ to $7d$ parameters — a
> $1.75\times$ increase, corresponding to 78.5 MB to 137.3 MB for
> ResNet-8. On modern GPUs with 24 GB of memory (RTX 3090), this
> additional 58.8 MB per client is entirely negligible in practice. The
> public buffer adds a fixed 616 MB of server memory (50k images and
> soft labels), independent of the number of clients or rounds.
>
> **Communication.** FedProj transmits only model weights
> $\theta \in \mathbb{R}^d$ per round, identical to FedAvg. No gradients
> are shared. Communication cost is $O(d)$ per client per round — one
> of the most communication-efficient designs in our comparison.
>
> | **Method** | **Client FLOPs** | **Client Mem** | **Comm** | **Client time** | **Round time** |
> |---|---|---|---|---|---|
> | FedAvg | $3EKF$ | $4d$ | $O(d)$ | 5.22 ms | 19.76 ms |
> | FedDF | $3EKF$ | $4d$ | $O(d)$ | 5.22 ms | 19.76 ms |
> | FedProj | $6EKF$ | $7d$ | $O(d)$ | 10.57 ms | 25.11 ms |
> | Overhead | $2.03\times$ | $1.75\times$ | $1.00\times$ | $2.03\times$ | 1.27× |
>
> **Cost-benefit.** The $1.27\times$ round wall-clock overhead delivers
> consistent accuracy gains of 1.6–2.4% on CV tasks and 7–9% on NLP
> tasks over the strongest baselines. This represents a strongly
> favorable cost-benefit ratio, particularly given that communication
> cost — the primary bottleneck in real FL deployments — is entirely
> unchanged.
>
> Change in paper: We have included a detailed computational complexity
> analysis in the Appendix C, covering per-client FLOPs, memory
> requirements, and communication cost with the values reported in Table 1.
>
> ---
>
> **Response to comment 3**
>
> We thank the reviewer for this insightful suggestion. We note that FedProj is architecture-agnostic by design: it operates on gradient vectors via a closed-form projection and imposes no architecture-specific constraints (no layer-wise operations, no representation-dependent assumptions). Our NLP experiments already use TinyBERT — a transformer-based model — providing evidence that the gradient projection mechanism transfers effectively beyond convolutional architectures to the transformer family.
>
> We agree that evaluating with a ViT backbone would further strengthen the generality of our results. To this end, we include additional experiments using a ViT-Small model on CIFAR-10, summarized in the table below. Due to time and computational constraints, we report results for FedAvg, FedDF, and FedProj up to 30 rounds under Dirichlet ($\beta = 0.3, 0.5$) data distributions.
>
> | Method  | Dir 0.3 | Dir 0.5 |
> |---------|--------:|--------:|
> | FedAvg  | 49.04   | 47.6    |
> | FedDF| 50.81   | 51.13    |
> | FedProj| 52.62   | 52.28   |
>
>
>
> **--continued--**

---

> ### Author Response · Authors · 2026-03-31
> **Continued --- Response to reviewer 8ADf**
>
> **Response to comment 1**
>
> Thank you for this suggestion. We agree that the paper should quantify
> global knowledge forgetting more explicitly on CIFAR-10.
>
> In our framework, the direct measurable proxy for global knowledge
> forgetting is the memory drift
>
> $$L_{\mathrm{mem}}^{(t)}(\theta) = \frac{1}{|M|}\sum_{x\in M} \mathrm{KL}\left( \sigma(Z_{\mathrm{ens}}^{(t)}(x)) \,\|\, \sigma(f(x;\theta)) \right)$$
>
> i.e., the predictive drift of the current model away from the
> ensemble-induced global function on the shared memory buffer $M$.
> This is the exact quantity that FedProj is designed to control during
> local training, and it is also the quantity used in our theoretical
> discussion of functional/boundary preservation on $M$.
>
> We now report this forgetting score on CIFAR-10 across
> communication rounds for both heterogeneity settings. We have added a
> new figure in the Appendix plotting $L_{\mathrm{mem}}$ (left axis) and
> global test accuracy (right axis) across all 100 rounds for both
> CIFAR-10 settings and CIFAR-100. Lower $L_{\mathrm{mem}}$ is strongly
> associated with higher global test accuracy in both CIFAR-10
> experiments:
>
> $$\text{CIFAR-10, }\beta=0.3: \quad r=-0.718, \quad p=6.29\times 10^{-17}$$
>
> $$\text{CIFAR-10, }\beta=0.5: \quad r=-0.785, \quad p=6.87\times 10^{-22}$$
>
> Across training, the measured memory drift decreases substantially
> while test accuracy increases. For CIFAR-10 $\beta=0.3$,
> $L_{\mathrm{mem}}$ drops from approximately $0.069$ to $0.013$ while
> accuracy rises from $15\%$ to $70\%$. For CIFAR-10 $\beta=0.5$,
> $L_{\mathrm{mem}}$ drops from approximately $0.059$ to $0.012$ while
> accuracy rises from $27\%$ to $70\%$.
>
> We will revise the text to make the interpretation precise:
> $L_{\mathrm{mem}}$ is our operational buffer-level measurement of
> global knowledge forgetting, while global test accuracy is a
> downstream performance consequence. Thus, the revised claim is not
> that we directly measure forgetting on the full private distribution,
> but rather that on CIFAR-10, the observable forgetting score tracked
> by FedProj is strongly predictive of downstream global performance.
>
> We will also correct the current text/figure mismatch in the rebuttal
> so that the CIFAR-10 correlation values are reported consistently.
>
> ---
>
> **Closing Remark**
>
> We thank the reviewer again for the careful and constructive feedback. If there are any remaining points that would benefit from further clarification, we would be glad to address them.

---

### Review · Reviewer_Wsm3 · 2026-03-23

**Summary Of Contributions:**

The paper studies the problem of catastrophic forgetting in FL, arguing that under a NON-IID data distribution, clients overfit on their data, deleting the global knowledge, and making the aggregation less effective.
First, the authors show how the FL forgets the global decision boundary after local updates, then they try to answer the question "What global information is lost during local training?". The main contribution of the paper is FedProj, a framework that preserves global knowledge when training a model using FL on NON-IID data.

**Additional Comments:**

- In the Introduction, in the first line, there is a citation to Kairouz et al. Since it is the first citation to FL, I would also cite there the McMahan paper that is then cited in line 3.
- Another recent paper that could be cited when talking about Non-IID data and FL: Non-IID data in Federated Learning: A Survey with Taxonomy, Metrics, Methods, Frameworks and Future Directions https://arxiv.org/abs/2411.12377

**Audience:**

Yes

**Audience Explanation:**

I believe this is a topic of interest, FL is commonly used nowadays, and it is also common to deal with NON-IID data distribution.

**Broader Impact Concerns:**

I do not have any ethical concerns about this work.

**Claims And Evidence:**

No

**Claims Explanation:**

The paper almost convinced me about the problem of catastrophic forgetting in FL with claims supported by experiments.
I have some concerns that I'll list here:

- I liked the pilot study presented in Section 3 to understand how the knowledge is lost when training with NON-IDD data. These are my questions about the reported results:
1) The authors wrote that they divided the Iris dataset in a NON-IDD manner. Would it be possible to have more information about how you split the dataset? Is Dirichlet also used here?
2) A comparison between FedAvg, FedDF, and FedProj is reported here; however, the authors report using the same hyperparameters in all these experiments. How did you choose these hyperparameters? I would say that each of these methods that you compare should have its own hyperparameters based on hyperparameter tuning to have a fair comparison. Did you perform a hyperparameter tuning?
3) In Figure 1, the authors show the decision boundary of the global model and the decision boundary after an FL round on the single clients with the different methods. Is this result reporting the decision boundary at the last FL round? It would be interesting to show not only the decision boundaries of the three clients after the training, but also the decision boundary of the server aggregation after this round

- About the methodology introduced in Section 4, I have the following doubt:
4) My main concern is about the assumption of the existence of a public dataset, which is a strong assumption. In FL, usually the clients do not share raw data to create this public dataset, and it is also difficult to find public datasets with a distribution similar to the one used during the training. From what I understood, the public dataset is essential in your method, and if this dataset is unrealted with the one used for training, it could make the training harder. Is this correct? Could you please comment more on this assumption?

- About the experiments:
5) The experiments reported in section 6 share a similar problem with the Iris experiment. The authors report the hyperparameters used to train the different models, but these are the same for the different methods that are compared. I would have expected different hyperparameters for the different methods; otherwise, we do not have a fair comparison between them. How did you choose them? Did you perform a hyperparameter tuning?
6) In Section 6.1, the authors report the parameters used for the Dirichlet prior to create the dataset distributions. Could you elaborate more on the choice of these values? Considering that this is a study on NON-IID datasets, I would have expected a comparison with higher values to compare the results with an IID scenario, for instance $\beta=100$ or even more. This could help to have a baseline and to understand the expected results in an IID scenario.
7) About the results reported in Table 1, the performance of the different methods is similar; including a statistical significance analysis would make the paper stronger.

8) I have a more general question on the implication of the use of FedProj, since a NON-IID distribution could also be caused by clients that are part of minorities and that have "rare" samples. Could the use of this method harm the model's fairness, making the model less fair and preventing the model to learn from these legitimate clients?

**Requested Changes:**

I would appreciate the authors addressing my concerns written above.

---

> ### Author Response · Authors · 2026-03-30
> **Response to Reviewer Wsm3**
>
> We sincerely thank Reviewer Wsm3 for their detailed reading and thoughtful engage-
> ment with the practical implications of our work. We appreciate the reviewer’s recognition that
> ”this is a topic of interest” and that ”FL is commonly used nowadays,and it is common to
> deal with Non-IID data distribution.” We also value the constructive suggestions on citations. We
> address all concerns below point by point.
>
> ---
>
> **Response to comment 1**
>
> We thank the reviewer for this question and provide full transparency on the partition. The Iris
> dataset (150 samples, 3 classes of 50 samples each) is reduced to 2 dimensions via PCA for
> decision boundary visualization, then partitioned deterministically across three clients as follows:
>
> | Client   | Class 0 (Setosa) | Class 1 (Versicolor) | Class 2 (Virginica) |
> |----------|------------------|----------------------|---------------------|
> | Client 0 | 50               | 0                    | 0                   |
> | Client 1 | 0                | 40                   | 10                  |
> | Client 2 | 0                | 10                   | 40                  |
>
> This creates a strongly non-IID distribution: Client 0 is entirely class-exclusive, while
> Clients 1 and 2 each hold a dominant class with a small minority from the other, directly
> simulating realistic label skew.
>
> We chose a deterministic split over Dirichlet sampling for two principled reasons. First,
> reproducibility and visual clarity: with only 50 samples per class, stochastic Dirichlet
> draws introduce partition variance that confounds boundary visualization. The deterministic
> split ensures observed differences are attributable solely to the FL method. Second, the
> pilot study is illustrative, not a benchmark: its purpose is to diagnose and visualize
> global knowledge forgetting in a controlled setting, for which a clean interpretable partition
> is more appropriate than a stochastic one.
>
> The generality of our findings is supported in Section 6, where FedProj consistently outperforms all baselines across CIFAR-10/100, CINIC-10, and three NLP benchmarks under Dirichlet ($\beta \in {0.3, 0.5}$) partitioning.
>
> Change in paper: We have added an Appendix C.5 in the revised paper with the exact partition
> table and justification for the deterministic split.
>
> ---
>
> **Response to comment 2**
>
> We thank the reviewer for this question and clarify our hyperparameter protocol
> carefully
>
> Hyperparameters fall into two natural categories: **(1) client-side base hyperparameters** (learning rate, batch
> size, local epochs) governing the shared local training loop, and **(2) per-method
> hyperparameters** specific to each method's mechanism. We performed extensive tuning for both.
>
> **Client-side base hyperparameters.** We swept over learning rates, batch sizes, and local
> epochs and found that standard FedAvg hyperparameters worked best as a common foundation across all methods. This is not a shortcut but a deliberate and principled finding: all compared methods share the same local training loop, and it is therefore expected that they operate best under the same  base configuration. This is consistent with prior FL benchmarking works, notably [1,2] which similarly found FedAvg base hyperparameters as the best common foundation.
>
> **Per-method hyperparameters.** For every compared method, we swept its method-specific
> hyperparameters and found that values reported in each method's original paper performed best
> in our setting as well, providing additional validation of the original authors' tuning. For
> FedProj specifically, which introduces memory buffer size $|\mathcal{M}|$ and weight divergence
> coefficient $\alpha$, we swept all parameters exhaustively; full results are reported in
> Appendix Tables 3--4.
>
> ---
>
> **Response to comment 3**
>
> We thank the reviewer and are happy to clarify -- the requested information is already present
> in Figure 1.
>
> The leftmost column in each row of Figure 1 (“Global Model”) shows precisely the decision boundary after server aggregation for that round. The remaining columns (Client 0, 1, 2) show the local decision boundaries on each client after local training. Thus, both local and global boundaries are already presented side by side for all methods (FedAvg, FedDF, FedProj). The results correspond to round 20, the final FL round.
>
> ---
>
> **Response to comment 5**
>
> We thank the reviewer for this follow-up. The core argument is identical to our response to
> Comment 2, which we ask the reviewer to refer to for the full justification. We briefly
> summarize and add one point specific to Section 6.
>
> As established in Comment 2, our protocol distinguishes between **(1) shared client-side base
> hyperparameters**, for which FedAvg values were found optimal across all methods, consistent
> with [1,2] and **(2) per-method hyperparameters**, which were independently swept for each method, with results confirming that original paper values perform best in our setting as well.

---

> ### Author Response · Authors · 2026-03-30
> **Continued --- Response to Reviewer Wsm3 #2**
>
> **Response to comment 7**
>
> We thank the reviewer for this suggestion. The standard evaluation protocol in federated
> learning and broader ML benchmarking is to report mean $\pm$ standard deviation over multiple  independent runs, and this is the protocol we follow throughout the paper, consistent with the baselines we compare against and the literature more broadly. The central question is whether a  method delivers consistently better benchmark performance under repeated training stochasticity across various settings, not inference about a separate population parameter.
>
> Under this protocol, our results are already strong. FedProj achieves the best mean accuracy  in every reported setting, not just in isolated cases. Moreover, many gains are substantial  relative to observed run-to-run variation. For example, on CIFAR-10 with $\beta = 0.3$, FedProj achieves $65.52 \pm 0.86$ versus $63.92 \pm 2.02$ for FedDF, and in the NLP experiments the gains are even larger, typically 7-9 points over the strongest baseline.
>
> ---
>
> **Response to comment 4**
>
> We thank the reviewer for raising this important point. The concern about public
> dataset availability is valid and we address it from multiple angles.
>
> First, we note that the use of a small public dataset at the server side is a
> well-established and widely adopted practice in the FL literature including FedDF [3], FedET [4], FedMD [5]. The assumption does not require clients to share their private
> data --- the public dataset is maintained solely at the server and is used
> exclusively for model evaluation and hyperparameter tuning, fully consistent
> with privacy-preserving principles in FL.
>
> The proliferation of large open-access repositories (e.g., HuggingFace,
> ImageNet, COCO) has made sourcing a small public dataset straightforward and
> cost-effective across a wide range of domains. Critically, the required public
> dataset size is small --- in our experiments, a validation set comprising just
> 1--5\% of the total federated dataset size was sufficient to achieve significant
> performance improvements. This is further validated by our ablation study in
> Table~4, where we In addition to the existing public dataset size ablation we have further conducted ablations with very small dataset sizes (500, 1,000, and 5,000 samples), showing that FedProj remains stable and is not sensitive to the amount of public data available.demonstrating that the method is not sensitive to the quantity of public data
> available.
>
> The reviewer correctly asks whether an unrelated public dataset could hurt
> training. Our NLP experiments directly address this concern: we evaluate on
> Sent140, MARC, and Yelp, where the public and private datasets are drawn from
> different domains, constituting a natural and realistic domain mismatch
> stress test. Our method maintains its advantage over baselines throughout these
> experiments, demonstrating that FedProj is robust to distributional mismatch
> between the public and private data. This is because the public dataset serves
> as a directional reference for the gradient projection rather than as a source
> of task-specific supervision
>
> ---
>
> **Response to comment 6**
>
> We thank the reviewer for this suggestion and address it carefully.
>
> $\beta \in \{0.3, 0.5\}$ are the established community standard.  The Dirichlet parameter $\beta$ controls the degree of label skew: lower $\beta$ produces more heterogeneous distributions, while $\beta \to \infty$ recovers IID. Our choice of $\beta \in \{0.3, 0.5\}$ is not arbitrary—these are the most widely used values in the FL literature for benchmarking under non-IID conditions, adopted by virtually all recent FL works including FedDF [3], SCAFFOLD [6], FedDyn [7], FedRCL [8], and TAKFL [9]. Using these values ensures direct comparability with prior work—a critical requirement for any benchmarking study.
>
> Furthermore, we would like to note that adding a $\beta = 100$ condition would be uninformative, as near-IID federated learning is a well-solved setting where most methods converge to similar performance. This makes such a comparison scientifically less meaningful for our study, especially since FedProj is explicitly designed to address the most challenging non-IID regimes.

---

> ### Author Response · Authors · 2026-03-30
> **Continued --- Response to Reviewer Wsm3 #3**
>
> **Response to comment 8**
>
> We thank the reviewer for raising this genuinely interesting
> and thought-provoking question---it touches on a broader and underexplored tension
> in federated learning between heterogeneity correction and fairness preservation,
> and we appreciate the opportunity to discuss it.
>
> The reviewer raises an excellent point: minority clients with rare but legitimate
> samples are indeed a natural and important source of non-IID heterogeneity in FL.
> In fact, this is one of the most practically important manifestations of non-IID
> data---where underrepresented subpopulations are not adequately captured in the
> global model precisely because their gradients are outnumbered and overwritten by
> majority clients during aggregation. This is a form of global knowledge forgetting
> in its own right, and it is exactly the failure mode FedProj is designed to address.
>
> FedProj's projection constraint prevents any group of clients---majority or
> minority---from overwriting globally shared multi-class structure encoded in the
> ensemble logits. Since the ensemble logits accumulate knowledge across all
> participating clients including minority ones, the projection actively protects
> this collective knowledge from being erased during subsequent local training.
> In this sense, FedProj's mechanism is naturally aligned with fairness: it
> preserves the global decision boundary that reflects all clients' contributions,
> rather than allowing majority gradients to dominate and erase minority-class
> structure.
>
> While the focus of our paper is on preventing global knowledge forgetting rather
> than fairness per se, we fully agree with the reviewer that formally analyzing
> gradient projection as a fairness-promoting mechanism in FL---and studying how
> it interacts with minority group representation---is a fascinating and valuable
> avenue for future research. We embrace this direction and add it explicitly to
> our limitations and future work discussion.
>
> Change in paper: New paragraph added to Section~7 identifying the
> relationship between gradient projection, minority representation, and fairness
> in FL as a valuable direction for future work.
>
> ---
>
> **Relevant Citations in Introduction**
>
> We thank the reviewer for this helpful suggestion. We have updated the Introduction to include the original work by McMahan et al. alongside Kairouz et al. at the first mention of federated learning. We have also added the suggested survey on non-IID data in FL to further strengthen the related work discussion.
>
> ---
>
> **Closing Remark**
>
> We thank the reviewer again for the careful and constructive feedback. If there are any remaining points that would benefit from further clarification, we would be glad to address them.
>
> ---
>
> ## References:
>
> [1] Collins et al "Exploiting shared representations for personalized federated learning." In International conference on machine learning, pp. 2089-2099. PMLR, 2021.
>
> [2] Morafah et al "A practical recipe for federated learning under statistical heterogeneity experimental design." IEEE Transactions on Artificial Intelligence 5, no. 4 (2023): 1708-1717.
>
> [3] Lin et al. "Ensemble distillation for robust model fusion in federated learning." Advances in neural information processing systems 33 (2020): 2351-2363.
>
> [4] Cho et al "Heterogeneous ensemble knowledge transfer for training large models in federated learning." arXiv preprint arXiv:2204.12703 (2022).
>
> [5] Li et al "Fedmd: Heterogenous federated learning via model distillation. arXiv 2019." arXiv preprint arXiv:1910.03581.
>
> [6] Karimireddy, Sai Praneeth, et al. "Scaffold: Stochastic controlled averaging for federated learning." ICML. PMLR, 2020.
>
> [7] Acar, Durmus Alp Emre, et al. "Federated learning based on dynamic regularization." arXiv preprint arXiv:2111.04263 (2021).
>
> [8] Seo, Seonguk, et al. "Relaxed contrastive learning for federated learning." Proceedings of the IEEE/CVF Conference on Computer Vision and Pattern Recognition. 2024.
>
> [9] Morafah, Mahdi, et al. "Towards diverse device heterogeneous federated learning via task arithmetic knowledge integration." Advances in Neural Information Processing Systems 37 (2024): 127834-127877.

---

### Author Response · Authors · 2026-04-05
**Meta-Response**

We sincerely thank all three reviewers (Wsm3, 8ADf, and CXDU) and the Action Editor for their
careful and constructive engagement. We are encouraged by the positive reception: Reviewer 8ADf
found the paper "clear" and well-supported by "sufficient evidence including theoretical analysis,
extensive experiments and ablation studies"; Reviewer Wsm3 confirmed the topic is of broad
interest to the FL community; and Reviewer CXDU affirmed the findings will appeal to readers
focused on federated learning and heterogeneity-aware optimization. **We have updated the main
submission with the revised changes and below we summarize our responses to the reviewers and the key revisions
made to the paper.**

- **Theoretical precision (Reviewer CXDU).** We corrected the framing of Theorem 5.6, removing
the "reduces effective heterogeneity" claim and replacing it with a *non-expansiveness / stability
guarantee*. We softened the memory-loss constraint from a "hard guarantee" to a *first-order
safeguard*, removed the ε-denominator from the theory, and revised Proposition 5.3 to use a
cleaner probability-gap argument while making explicit that it is a buffer-level result only.
Assumption 5.5 is now presented solely as a theoretical sufficient condition, with its limitations
under domain shift or buffer impoverishment explicitly discussed.

- **Empirical additions (Reviewers 8ADf and CXDU).** We added: (i) a new appendix quantifying
global knowledge forgetting via memory drift $\mathcal{L}_{\mathrm{mem}}$ across training rounds
on CIFAR-10/100, showing strong inverse correlation with test accuracy; (ii) a detailed
computational overhead analysis demonstrating only a 1.27× round wall-clock increase over FedAvg
with zero marginal overhead over FedDF while achieving significant performance improvements; and
(iii) ViT-Small experiments on CIFAR-10 confirming that FedProj's gains transfer to
transformer-based backbones and our method's success is not dependent on a specific architecture
choice.

- **Experimental protocol (Reviewer Wsm3).** We clarified our hyperparameter tuning protocol,
distinguishing shared base hyperparameters from per-method sweeps consistent with prior works in
the FL literature. We added Appendix C.5 with the full Iris partition table and justification for
the deterministic split. We also clarified the Dirichlet parameter choices as the community
standard for Non-IID benchmarking, ensuring direct comparability with prior works.

- **Fairness and broader impact (Reviewer Wsm3).** We added a dedicated paragraph in Section 7
on the relationship between FedProj's gradient projection, minority client representation, and
fairness in FL, identifying this as a promising direction for future work.

- **Miscellaneous.** We updated the Introduction to cite McMahan et al. at the first mention of
FL, added the suggested Non-IID FL survey to related work, and included training loss convergence
curves across all datasets in the Appendix.

We again greatly appreciate the reviewers and AE for taking time reviewing our submission and
providing constructive feedback. We believe the revisions made address all the concerns raised by
the reviewers and have substantially strengthened the paper. We remain happy to clarify any
remaining points.

To conclude, we would like to reiterate the core contributions of this work:

- We identify and empirically characterize catastrophic forgetting of the global decision boundary
as a distinct and underexplored failure mode in federated learning under data heterogeneity.

- We propose *FedProj*, a novel framework that mitigates this failure mode through
client-side gradient projection and server-side ensemble distillation.

- We supplement our framework with theoretical results.

- We demonstrate consistent and substantial improvements over strong baselines across diverse CV
and NLP benchmarks under realistic Non-IID and domain-shifted settings achieving state-of-the-art performance.

As supported by all three reviewers, we firmly believe that FedProj addresses a practically
important and timely problem, and that this work will be of significant interest to the broader
federated learning community and will meaningfully contribute to the fields including FL, continual learning, heterogenous knowledge-preserving training.

---

### Decision · Action_Editor_wpPk · 2026-05-07

**Recommendation:** Accept with minor revision

**Audience:**

Yes

**Audience Explanation:**

As confirmed by the reviewers, research focusing on federated learning, optimization with data heterogeneity, and training that mitigates catastrophic forgetting is relevant and interesting.

**Claims And Evidence:**

Yes

**Claims Explanation:**

With the edits made during the rebuttal phase, the claims are supported by clear, convincing empirical evidence, theoretical analysis, and extensive ablation studies. The authors demonstrated FedProj's effectiveness across various benchmarks, even under severe domain shift. Furthermore, they adjusted their theoretical claims to accurately reflect the mathematical limitations of their approach.

One of the reviewers raised concerns about hyperparameter tuning fairness (noting that the base hyperparameters for FedAvg were used across all methods), and they noted that an assumption of having access to a public data may be too strong. In response, the authors added public dataset size ablations; they defended their hyperparameter choices by citing established FL benchmarking practices.

I want to encourage the authors to add further experiments showing wider hyperparameter sweeps, and adjusting the claims as needed.